# LieStoNet: Learning Lie Symmetries from Spatiotemporal Data for Stochastic Dynamical Systems

Shida Liu [* 1]   Abhishek Gupta [* 2]   Sumit Sinha [* 1]   L. Mahadevan [3]

## Abstract

Symmetry is central to modern machine learning and physics: invariances and equivariances improve sample efficiency, robustness, and out-of-distribution generalization, while symmetry principles guide scientific modeling. Yet for stochastic dynamical systems the relevant continuous symmetries are rarely known, and symmetry discovery for SDEs has remained essentially unexplored. We introduce *LieStoNet*, an end-to-end, *template-free* framework for discovering Lie-point symmetries of SDEs directly from spatiotemporal trajectories, without prespecifying symmetry groups, templates, or canonical coordinates. Building on the seminal SDE Lie-symmetry theory of Gaeta and Quintero (1999), which formalizes Lie-point SDE symmetries and their relation to Fokker-Planck symmetries, LieStoNet learns neural surrogates for drift and diffusion from increments, then learns projectable generators by enforcing the SDE determining equations, separately regularizing for closure under Lie brackets, adherence to the Lie algebra axioms (bilinearity, antisymmetry, Jacobi), and a non-redundant independent basis. The surrogate also defines an associated Fokker-Planck equation, enabling optional discovery of its Lie-point symmetries in parallel. Across multiple canonical SDEs with known analytic symmetries, LieStoNet recovers generators consistent with the ground-truth symmetry algebra, providing interpretable symmetry discovery for noisy dynamics. Code is available at this link.

*Equal contribution   [1]School of Engineering and Applied Sciences (SEAS), Harvard University, Cambridge, MA, USA. [2]Tracelink, Wilmington, MA, USA. [3]SEAS, Physics and OEB, Harvard University, Cambridge, MA, USA. Correspondence to: Sumit Sinha <ssinha@g.harvard.edu>, L. Mahadevan <lmahadev@g.harvard.edu>.

*Proceedings of the 43rd International Conference on Machine Learning*, Seoul, South Korea. PMLR 306, 2026. Copyright 2026 by the author(s).

## 1. Introduction

Symmetry is a foundational idea in science: it captures the notion that certain transformations of a system leave its behavior unchanged. In physics, symmetry principles have repeatedly guided the construction of successful theories and helped explain why diverse phenomena can be described by the same underlying laws (Gross, 1995; Gross and Wilczek, 1973). In machine learning, symmetry plays a similarly structural role. When a task is known to be insensitive to specific transformations (e.g., rotations, translations, permutations), incorporating the corresponding invariance or equivariance into models can substantially improve sample efficiency, robustness, and out-of-distribution generalization. This has motivated a broad line of work on symmetry-respecting architectures and learning pipelines, including group-equivariant convolutional networks (Cohen and Welling, 2016; Cohen et al., 2019), rotation/translation-equivariant attention and geometric deep learning methods (Fuchs et al., 2020; Thomas et al., 2018; Satorras et al., 2021; Kondor et al., 2018), and equivariant generative models for scientific simulation (Kanwar et al., 2020; Boyda et al., 2021). More broadly, symmetry is a key ingredient in physics-guided learning for dynamical systems (Wang and Yu, 2021), and can complement approaches that learn conserved quantities or structured dynamics to enhance interpretability and generalization (Greydanus et al., 2019; Alet et al., 2021).

Despite these advances, the most impactful symmetries are often *unknown* in the settings where we would most like to exploit them. Real-world data may come from partially observed systems, complex coordinate representations, or measurements that mix latent variables, so the relevant transformations are not obvious. This motivates *symmetry discovery*: learning the symmetry structure directly from data, rather than prescribing it by hand. Recent work has advanced symmetry discovery for deterministic dynamical systems, ranging from methods that infer invariances from learned representations/predictors to approaches that explicitly learn continuous transformations or generators, with downstream use in scientific modeling such as governing-equation and variable discovery (Benton et al., 2020; Liu and Tegmark, 2022; Yang et al., 2024a; Forestano et al.,

2023; Ko et al., 2024; Shaw et al., 2024; Yang et al., 2024b; Mohapatra et al., 2025). Collectively, these results position symmetry discovery as a practical route to interpretable, data-efficient learning in deterministic settings.

In this paper we move to the stochastic regime. Many systems of interest are inherently stochastic - due to unresolved degrees of freedom, environmental variability, or deliberate modeling of uncertainty - and stochastic differential equations (SDEs) provide a standard, flexible language for such dynamics. SDEs arise broadly across machine learning and the sciences, from diffusion-based generative modeling and Langevin-type sampling to canonical stochastic models in quantitative finance, molecular/biophysical dynamics, neuroscience, and climate/geophysical systems. Yet, while symmetry discovery for deterministic equations has rapidly evolved, *symmetry discovery for SDEs remains essentially unexplored: to the best of our knowledge, there is no existing symmetry-discovery pipeline for SDEs at all*. We close this gap with **LieStoNet**, an end-to-end framework that learns continuous SDE symmetries directly from trajectory data without prespecifying the symmetry group. In particular, "symmetry" may mean preserving sample-path behavior, preserving the evolution of probability distributions, or preserving derived deterministic descriptions of the system. A central derived description is the Fokker–Planck equation, which governs how the state's probability density evolves over time. Because it is deterministic, the Fokker–Planck viewpoint is attractive for analysis and validation, but it does not automatically settle what it means to be a symmetry of the underlying stochastic dynamics. A principled approach must therefore ground the learning objective in the correct stochastic definition while still leveraging the Fokker–Planck perspective when useful.

**Lie-point symmetry theory for SDEs.** Our approach is grounded in the Lie-point (continuous) symmetry theory for SDEs of Gaeta and Rodríguez Quintero (Gaeta and Quintero, 1999). This theory formalizes when a smooth, continuously-parameterized transformation maps an SDE to an equivalent SDE while respecting its stochastic structure, and it clarifies how SDE symmetries relate to (yet need not coincide with) symmetries of the associated Fokker–Planck equation. While it provides the correct conceptual target, it is not itself a data-driven discovery method. Our goal is to translate these principles into LieStoNet, a modern ML pipeline that discovers such symmetries directly from trajectory data.

**Contributions.** We summarize our main contributions as follows:

- **SDE symmetry discovery.** We introduce LieStoNet, to the best of our knowledge the first end-to-end ML framework for discovering continuous (Lie-point) symmetries of stochastic dynamical systems, addressing

a gap in the symmetry-discovery literature that has largely focused on deterministic equations.

- **Template-free discovery pipeline.** LieStoNet does not prespecify a symmetry group, canonical coordinates, symbolic library, or generator template. Instead, it learns symmetry structure directly from spatiotemporal data while enforcing the stochastic symmetry conditions of Gaeta and Quintero (1999). The method is not assumption-free: it operates within the class of projectable Lie-point SDE symmetries, while broader stochastic symmetry classes, including random Lie-point symmetries, have also been studied (Gaeta and Spadaro, 2017). The neural parameterization and optimization procedure also introduce inductive biases. We therefore use "template-free" to mean free of pre-specified symmetry templates within this projectable SDE symmetry class.

- **Connecting SDE and Fokker–Planck viewpoints.** By learning a neural surrogate of the underlying SDE, our framework also induces a surrogate Fokker–Planck description for probability density evolution. We show symmetry discovery is possible in this associated deterministic view as well; nevertheless, since the Fokker–Planck equation does not capture all subtleties of stochastic symmetry, we emphasize discovery and validation at the SDE level.

**Related works.** Our work is most closely connected to recent machine-learning approaches that aim to discover symmetries from observations, particularly in deterministic settings. This includes methods that infer symmetries through invariance patterns in learned representations or predictors (Benton et al., 2020; Liu and Tegmark, 2022; Yang et al., 2024a), adversarial and generative formulations (Yang et al., 2023), explicit generator-style learning of continuous symmetries (Forestano et al., 2023; Ko et al., 2024; Shaw et al., 2024), and data-driven Lie-point symmetry detection for deterministic continuous dynamical systems (Gabel et al., 2024). Related deterministic work has also used known Lie-point symmetries for data augmentation and equivariance in neural PDE solvers (Brandstetter et al., 2022). Other recent approaches go beyond restricted transformation classes (Shaw et al., 2024; Hu et al., 2025; Shaw et al., 2025). There is also growing interest in discovering discrete symmetry groups (Calvo-Barlés et al., 2024; 2025) and in identifying local or approximate symmetries in high-dimensional systems (Bhat et al., 2025). Finally, symmetry has been used as a structural prior for downstream scientific modeling tasks, including governing-equation and variable discovery as well as scientific representation learning (Yang et al., 2024b; Mohapatra et al., 2025; Wang et al., 2020).

| Paper | Generator type | | Lie algebra? | IIC? | Dynamics type | |
|---|---|---|---|---|---|---|
| Ko et al. (Ko et al., 2024) | VF ■ | M □ | □ | □ | STO □ | DET ■ |
| Forestano et al. (Forestano et al., 2023) | VF □ | M ■ | ■ | □ | STO □ | DET ■ |
| LieGAN (Yang et al., 2023) | VF □ | M ■ | ■ | □ | STO □ | DET ■ |
| Augerino (Benton et al., 2020) | VF □ | M ■ | □ | □ | STO □ | DET ■ |
| LaLiGAN (Yang et al., 2024a) | VF □ | M ■ | ■ | □ | STO □ | DET ■ |
| Shaw et al. (Shaw et al., 2024) | VF ■ | M □ | □ | □ | STO □ | DET ■ |
| LieNLSD (Hu et al., 2025) | VF ■ | M □ | □ | ■ | STO □ | DET ■ |
| **LieStoNet (Ours)** | VF ■ | M □ | ■ | ■ | STO ■ | DET □ |

*Table 1.* **Comparison with related symmetry discovery works.** ■ denotes yes and □ denotes no. "VF" = vector-field generator; "M" = matrix/linear parameterization. "STO" = stochastic; "DET" = deterministic. Columns indicate whether methods explicitly use Lie-algebra structure and an infinitesimal invariance condition (IIC). VF parameterizations can represent nonlinear, state-dependent symmetry transformations, whereas M parameterizations are typically limited to linear/affine actions. Lie-algebra regularization enforces coherent composition/closure among generators, stabilizes training, and promotes recovery of a complete basis spanning the symmetry space rather than a bag of unrelated invariances. IIC constraints provide a principled local criterion that can be enforced densely and differentiably, yielding more reliable and data-efficient symmetry learning than finite, sample-based checks alone.

## 2. Mathematical Preliminaries

In this section, we briefly review the mathematical preliminaries; see Appendix A for details.

### 2.1. Stochastic Differential Equations and Lie Algebraic Symmetries

We use the Itô convention to model stochastic dynamics. An Itô stochastic differential equation describes the evolution of a state $\mathbf{x}_t$ through two components: a drift term $f$, which captures the deterministic tendency of the dynamics, and a diffusion term $\sigma$, which captures random fluctuations driven by a Wiener process. In this work, symmetries of such systems are understood as transformations of time and state variables that preserve the stochastic model. Because SDE sample paths are random, this preservation is interpreted distributionally: a symmetry maps the process to another process with equivalent induced stochastic dynamics in the transformed coordinates. To describe continuous families of such transformations, we work infinitesimally. A smooth one-parameter transformation has an infinitesimal generator, written schematically as a vector field $X = (\tau, \xi)$, whose components describe the first-order changes in time and state. The full finite transformation can be recovered by integrating this vector field, and Itô's formula determines how the drift and diffusion coefficients transform under the resulting change of variables.

The infinitesimal symmetry generators naturally form a Lie algebra. A Lie algebra is a vector space equipped with a bilinear bracket operation that records how infinitesimal transformations fail to commute; the bracket is antisymmetric and satisfies the Jacobi identity. For vector-field generators, this bracket is the usual commutator $[X, Y] = XY - YX$, which measures the difference between applying two infinitesimal transformations in opposite orders. The set of SDE symmetry generators is closed under linear combinations and under this bracket, so learning several generators

amounts to learning a symmetry subspace together with its composition structure. For example, the span of the translation generator $\partial_x$ and the scaling generator $x\partial_x$ is closed because their bracket is again a generator in the same span, namely $[x\partial_x, \partial_x] = -\partial_x$. In what follows, our symmetry generators are represented as vector fields, and the Lie bracket provides the natural notion of closure under composition; the resulting Lie algebra is the main object we aim to discover and evaluate.

### 2.2. Determining Equations for Projectable Generators

For clarity we present the one-dimensional setting (higher dimensions similar), however the theory and our implementations extend to high-dimensional examples as well. A *projectable generator* is a Lie-point symmetry generator whose time component depends only on time, not on the state. We therefore restrict symmetry generators to the form

$$X = \tau(t)\, \partial_t + \xi(t, x)\, \partial_x. \tag{1}$$

This excludes state-dependent time reparameterizations while retaining the continuous symmetries considered in this work.

In deterministic ODE/PDE symmetry theory, a standard route to validating $X$ is the *infinitesimal invariance condition* (IIC): invariance under the generator's flow implies local differential constraints on $(\tau, \xi)$ (Olver, 1993). In the SDE setting, the analogous IIC must account for both drift and diffusion and the stochastic structure, yielding the *SDE determining equations* below.

$$dX_t = f(t, X_t)\, dt + \sigma(t, X_t)\, dW_t. \tag{2}$$

**Theorem 2.1** (SDE determining equations for projectable Lie-point symmetries (Gaeta and Quintero, 1999)). *Consider the Itô SDE* (2) *and a projectable generator $X$ as in* (1). *Then $X$ generates a (Lie-point) symmetry of* (2) *if and*

*only if $(\tau, \xi)$ satisfy, for all $(t, x)$,*

$$\xi_t + f\xi_x - \xi f_x - \partial_t(f\tau) + \tfrac{1}{2}\sigma^2 \xi_{xx} = 0,$$

$$\sigma\xi_x - \xi\sigma_x - \tau\sigma_t - \tfrac{1}{2}\sigma\tau_t = 0.$$

*Here subscripts denote partial derivatives (e.g., $\xi_t = \partial_t\xi$, $\xi_{xx} = \partial_{xx}\xi$), and $\partial_t(f\tau) = f_t\tau + f\tau_t$.*

We compute using these determining equations, the full symmetry Lie-algebras for four different SDEs in appendices J, K, L, M.

### 2.3. Fokker–Planck Symmetries

**Associated FP equation.** The SDE (2) induces a deterministic evolution equation for the one-point density $u(t, x)$ of $x_t$, namely the (forward) Fokker–Planck equation

$$u_t = -\partial_x\big(f(t, x)\, u\big) + \tfrac{1}{2}\partial_{xx}\big(\sigma^2(t, x)u\big). \quad (3)$$

Because (3) is deterministic and linear in $u$, it admits a well-developed Lie-point symmetry theory.

**FP symmetry generators.** A general Lie-point generator acting on $(t, x, u)$ has the form

$$Y = \tau(t, x, u)\partial_t + \xi(t, x, u)\partial_x + \phi(t, x, u)\partial_u. \quad (4)$$

One can prove that the linearity of FP implies that, the $u$-component of the generator must be affine in $u$: $\phi(t, x, u) = \alpha(t, x) + \beta(t, x)\, u$ (the $\alpha$-part corresponds to linear superposition) (see (Gaeta and Quintero, 1999)). The FP analogue of the IIC yields a system of determining equations for $(\tau, \xi, \alpha, \beta)$.

**Theorem 2.2** (FP determining equations for projectable Lie–point symmetries (Gaeta and Quintero, 1999))**.** *Consider the (one-dimensional) Fokker–Planck equation associated with the Itô SDE (2), written in the linear form*

$$u_t + A(t, x)u_{xx} + B(t, x)u_x + C(t, x)u = 0, \quad (5)$$

*where, for (3),*

$$A(t, x) = -\tfrac{1}{2}\sigma^2(t, x),\ B(t, x) = f(t, x) - \partial_x\big(\sigma^2(t, x)\big),$$

$$C(t, x) = \partial_x f(t, x) - \tfrac{1}{2}\partial_{xx}\big(\sigma^2(t, x)\big).$$

*Let $Y$ be a* projectable *Lie-point generator acting on $(t, x, u)$ with $u$-component affine in $u$:*

$$Y = \tau(t)\partial_t + \xi(t, x)\partial_x + \big(\alpha(t, x) + \beta(t, x)u\big)\partial_u. \quad (6)$$

*Then $Y$ is a Lie-point symmetry of (5) if and only if:*

*1. $\alpha(t, x)$ is any (fixed) solution of (5) (the linear superposition symmetry); and*

*2. $(\tau, \xi, \beta)$ satisfy the determining equations*

$$\partial_t(\tau A) + \xi\partial_x A - 2A\partial_x\xi = 0,$$

$$\partial_t(\tau B) - \partial_t\xi + B\,\partial_x\xi - \xi\partial_x B + 2A\partial_x\beta - A\partial_{xx}\xi = 0,$$

$$\partial_t(\tau C) + \partial_t\beta + A\partial_{xx}\beta + B\partial_x\beta + \xi\partial_x C = 0.$$

This theorem supplies an infinite family of symmetries: $\alpha(t, x)\partial_u$ for any solution $\alpha$ of the FP-equation. These are considered as trivial symmetries and are often quotiented-out when defining the Lie algebra $\mathfrak{g}_{FP}$ of Fokker-Planck symmetries. We present more details in the Appendix D. The analytic symmetry algebra generators for the FP equation associated with the Brownian motion $dx_t = \sigma_0 dW_t$ are presented in appendix N.

**SDE symmetries as a nested subalgebra of FP symmetries.** Additionally, SDE symmetries can be extended to FP symmetries which induces an injection of Lie algebras $\mathfrak{g}_{Ito} \hookrightarrow \mathfrak{g}_{FP}$ thus allowing us to think of SDE symmetries as a Lie-subalgebra of FP-symmetries. We provide more details along with a proof in Appendix E.

## 3. Symmetry Discovery Pipeline

On top of the above mathematical preliminaries, we present **LieStoNet**, an end-to-end pipeline for discovering a basis of infinitesimal generators that spans the Lie algebra of Lie-point symmetries of an *unknown* Itô SDE. We assume access only to discrete samples of trajectories (spatiotemporal data). In our experiments, these samples are produced by simulating analytic SDEs, after which we discard the governing equations and treat the data as observational data. *Fokker–Planck (FP) symmetries are deterministic*, so we only showcase FP-route symmetry discovery in Example 1 as a brief companion result, and otherwise keep the focus on *stochastic* (SDE-level) symmetry discovery to avoid expanding the scope.

| Ex. | Analytic Equations |
|---|---|
| 1 | $dx_t = 1\ dW_t$ |
| 2 | $dx_t = x_t\ dt + 1\ dW_t$ |
| 3 | $dx_t = y_t\ dt,\ dy_t = -1 \cdot y_t\ dt + \sqrt{2}dW$ |
| 4 | $dx_t = (1/x)dt + dW_1, dy_t = dt + dW_2$ |

*Table 2.* **Analytic equations for each example. The analytic ground-truth generators are derived in the appendices.**

### 3.1. Stage 1: Neural SDE Surrogate from Increments

**Data and surrogate.** We observe trajectories $\{x^{(k)}(t_n)\}_{k=1}^{K}$ on a grid $t_0 < t_1 < \cdots < t_N$ with

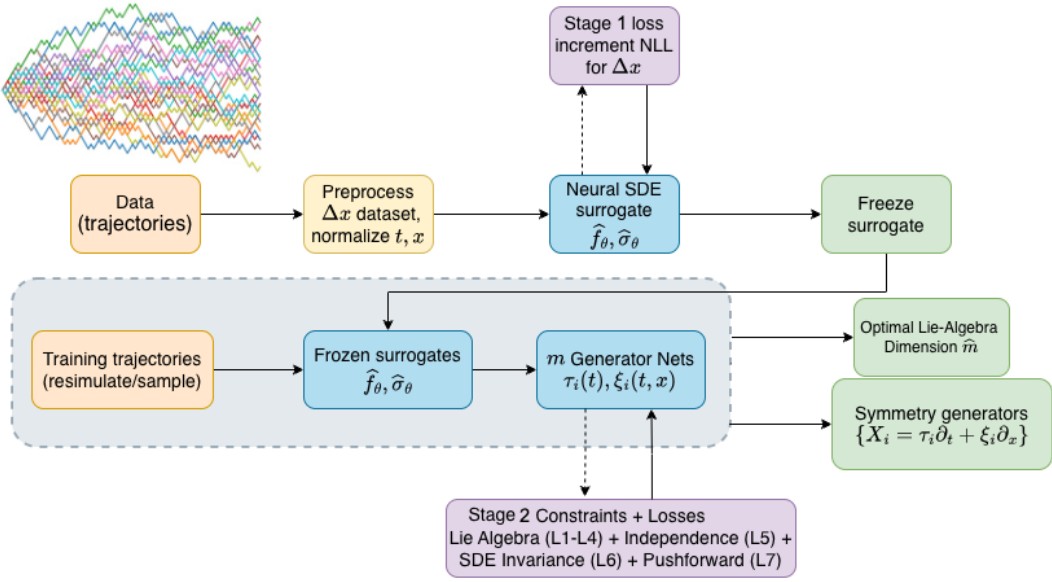

*Figure 1.* Pipeline for LieStoNet

$\Delta t = t_{n+1} - t_n$, and define increments

$$\Delta x_n^{(k)} := x^{(k)}(t_{n+1}) - x^{(k)}(t_n).$$

We fit a differentiable surrogate SDE

$$dx_t = f_\theta(t, x_t)\, dt + \sigma_\theta(t, x_t)\, dW_t, \qquad (7)$$

where $f_\theta$ and $\sigma_\theta > 0$ are parameterized by neural networks (details in the appendix P.0.1). The role of this surrogate is to provide smooth coefficient functions so that all derivatives required by the symmetry constraints in Theorem 2.1 and Appendix C can be computed via autodiff.

**Increment likelihood (local MLE).** Euler–Maruyama motivates the conditional approximation

$$\Delta x \mid (t_n, x_n) \approx \mathcal{N}(f_\theta(t_n, x_n)\Delta t, \ \sigma_\theta^2(t_n, x_n)\Delta t),$$

leading to the per-sample negative log-likelihood (up to an additive constant)

$$\ell_\theta = \frac{(\Delta x - f_\theta\,\Delta t)^2}{2\,\sigma_\theta^2\,\Delta t} + \frac{1}{2}\log(\sigma_\theta^2\,\Delta t). \qquad (8)$$

We minimize $L_{\mathrm{SDE}}(\theta) = \mathbb{E}[\ell_\theta] + \lambda_{\mathrm{reg}}\|\theta\|^2$ and then freeze $\theta$. In the remainder we write $f := f_\theta$ and $\sigma := \sigma_\theta$.

### 3.2. Stage 2: Generator Parameterization

**Projectable generators.** We learn $m$ projectable Lie-point generators of the form in Eq. (9)

$$X_i = \tau_i(t)\, \partial_t + \xi_i(t, x)\, \partial_x, \qquad i = 1, \dots, m, \quad (9)$$

where $\tau_i$ and $\xi_i$ are represented by multi-head MLPs producing $(\tau_1, \dots, \tau_m)$ and $(\xi_1, \dots, \xi_m)$. This enforces $\partial_x \tau_i \equiv 0$ by construction while allowing nonlinear, state-dependent symmetries through $\xi_i(t, x)$.

**Training points.** After fitting and freezing the Stage-1 surrogate $(f_\theta, \sigma_\theta)$, we generate a point cloud $\Omega$ for Stage-2 by simulating trajectories from the learned surrogate SDE and collecting the resulting space-time states $(t_n, x_n)$. All losses below are then computed as Monte Carlo averages over samples from $\Omega$ (and, when needed, over adjacent trajectory-time pairs).

### 3.3. Learning Objectives

LieStoNet trains $\{X_i\}_{i=1}^m$ using two families of losses. We first define *SDE-validity losses* that enforce Lie-point symmetry constraints for the surrogate SDE, then define *Lie-algebra structure losses* that regularize the learned generators to form a finite-dimensional Lie algebra under the Lie bracket. See Appendix C for detailed mathematical formulations.

**Lie-algebra structural losses**

- $L_1$ (**Lie-bracket closure + constancy, fixed**). Encourages $[X_i, X_j]$ to lie in $\mathrm{span}\{X_k\}$ (via projection) and the resulting structure coefficients $c_{ij}^k$ to be approximately constant over $(t, x)$.

- $L_2$ (**Jacobi identity**). Penalizes Jacobi violations so nested commutators compose consistently, i.e., the learned bracket behaves as a Lie bracket.

| Algorithm of LieStoNet | |
| --- | --- |
| **Input:** | Trajectories / spatiotemporal samples $D$; generator-count schedule $m = 1, 2, ...$ |
| **Output:** | Symmetry Lie-algebra dimension $m$, generators $\{X_k\}_{k=1}^K$, diagnostics |
| **Surrogate Learning** | Train a differentiable surrogates $f_\theta, \sigma_\theta$ on $D$. |
| **Generator Learning (fixed $m$)** | Parameterize $X_i$ for $i = 1, ..., m$. Optimize the composite objective: $\sum_{i=1}^7 w_i L_i$. |
| **Dimension Selection** | Increase $m$ and repeat generator learning; choose $\hat{m}$ at the first minimum of the closure loss $L_1(m)$ within the dimension bound. |
| **Report** | For the discovered dimension $m = \hat{m}$, compute span alignment with principal angles against analytic ground-truth generators (when available), evaluate/plot after-push residuals, and release $\{X_i\}$. |

*Table 3.* **Overview of LieStoNet**. The method learns vector-field generators by enforcing infinitesimal invariance and Lie-algebra closure constraints using learned drift and diffusion neural surrogates.

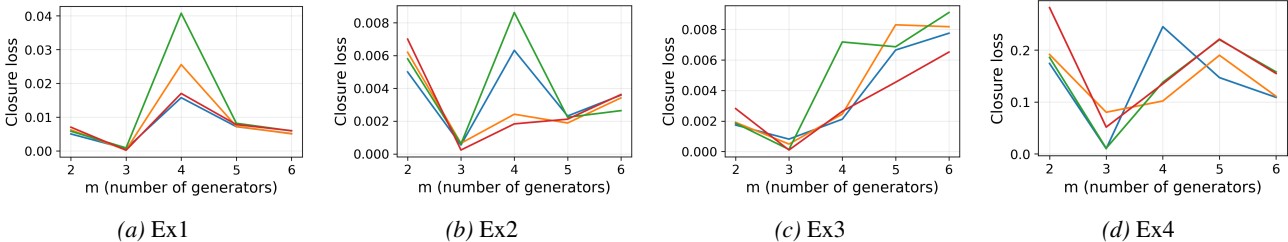

| *(a)* Ex1 | *(b)* Ex2 | *(c)* Ex3 | *(d)* Ex4 |

*Figure 2.* **Post-training Lie bracket closure loss $L_1$ vs. number of generators $m$ across four experiments.** Colors denote independent runs with identical settings (different random seeds), showing the minimum is consistently attained at the ground-truth m rather than a one-off.

- $L_3$ **(Skew-symmetry).** Enforces antisymmetry of the bracket: swapping generator order flips the commutator sign.

- $L_4$ **(Bilinearity).** Enforces linearity of the bracket in each argument so it distributes correctly over linear combinations.

- $L_5$ **(Functional independence).** Penalizes near-linear dependence of the learned fields over the domain to avoid redundant generators and ensure a truly $m$-dimensional span.

**SDE-validity losses**

- $L_6$ **(SDE determining-equation loss).** Enforces the Itô symmetry determining conditions (Theorem 2.1), making each learned generator compatible with the SDE drift and diffusion and thus a valid infinitesimal SDE symmetry.

- $L_7$ **(Prolonged pushforward residual; SDE-only, $\tau/\xi$).** Adds a finite-$\varepsilon$ check by flowing $(t, x)$ along the generator and comparing the pushed-forward surrogate coefficients to the surrogate coefficients evaluated at the pushed point, providing nonlocal validation beyond the infinitesimal conditions.

**FP losses** (When learning FP symmetries, we replace the SDE-validity losses with FP-specific losses while keeping

the same algebraic losses.)

- $L_8$ **(Fokker–Planck determining-equation loss).** Enforces Lie-point determining conditions for the induced Fokker–Planck PDE using generators $X_i = \tau_i(t)\partial_t + \xi_i(t, x)\partial_x + \beta_i(t, x)\,u\,\partial_u$, by minimizing the batch-averaged magnitude of the three FP determining-equation (2.2) residuals so each generator is a valid infinitesimal FP symmetry.

- $L_9$ **(Fokker–Planck after-flow / pushforward-on-$u$ loss).** Adds a finite-$\varepsilon$ check by flowing $(t, x, u)$ along the prolonged generator (including $\dot{u} = \beta_i(t, x)u$) and penalizing disagreement between the flowed value $u_\varepsilon$ and the FP surrogate evaluated at the flowed point $\hat{u}(t_\varepsilon, x_\varepsilon)$, i.e., approximately mapping FP solutions to FP solutions.

**Total objective.** Let $\boldsymbol{w} = (w_1, \ldots, w_7)$ denote nonnegative weights. We train generator parameters $\psi$ by minimizing

$$L_{\text{gen}}(\psi) = \sum_{j=1}^7 w_j\,L_j + w_{\text{wd}}\|\psi\|^2 \qquad (10)$$

where $w_{\text{wd}}$ represents the weight on weight decay of the generator parameters. Note that for the FP symmetry case, $L_6, L_7$ are replaced by $L_8, L_9$. See Appendix P for training details.

### 3.4. Determining the Lie algebra Dimension $m$

In practice, we follow the above symmetry generator training while sweeping through different candidates for the number of learned generators ($m = 1, 2, \ldots$) and record the post-training Lie bracket closure loss ($L_1$). The value of $m$ which minimizes $L_1$ is selected as the recovered dimension of the symmetry Lie algebra.

This sweep is finite in the SDE settings considered here because the admissible symmetry search space is a priori bounded: scalar Itô SDEs admit Lie algebras of dimension at most 3, while $n$-dimensional systems with full-rank diffusion have maximal dimension $n + 2$ (Kozlov, 2010a; 2011; 2010b). In contrast, deterministic differential equations may have arbitrarily large or infinite-dimensional symmetry algebras, with much larger finite bounds when they exist, e.g., $n^2 + 4n + 3$ for certain second-order ODE systems (González-López, 1988). Thus the Itô structure, diffusion determining equations, and noise irregularity make LieStoNet's dimension sweep a finite search over a theoretically justified range: one sweeps admissible $m$ and selects the minimizer of the post-training closure loss $L_1$. For genuinely high-dimensional systems, LieStoNet can also be applied after deriving a reduced effective SDE, e.g., via projection or Mori–Zwanzig-type coarse-graining.

## 4. Experiments: Evaluation Protocol

Element-wise comparison of learned symmetries is ill-posed because infinitely many bases can span the same Lie algebra; the basis-invariant object is therefore the *span* of the generators.

### 4.1. Span Alignment via Principal Angles (basis-invariant)

Concretely, we evaluate each set of generators over a shared $(t, x)$ region and view them as vectors in a common feature space; this yields a learned subspace and a ground-truth subspace. We then quantify how well these subspaces align using *principal angles*, which measure the smallest angular discrepancies between two subspaces: angles near $0°$ indicate that the learned generators span (up to basis change) the same symmetry directions as the analytic ones, while larger angles indicate missing or spurious directions. This metric is (i) basis-invariant, (ii) global over the evaluation region, and (iii) useful for diagnosing dimension mismatch (extra generators typically introduce directions that increase the worst-case misalignment) (see Appendix B).

### 4.2. Results

As shown in Figure 2, the post-training closure objective $L_1$ consistently attains its minimum at the analytic Lie-

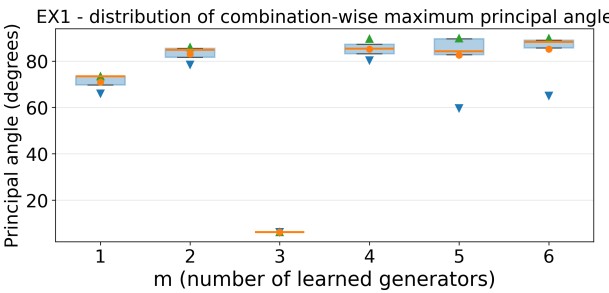

*(a)* Ex1

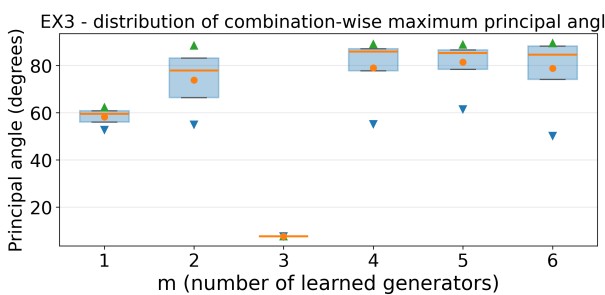

*(b)* Ex2

*(c)* Ex3

*(d)* Ex4

*Figure 3.* **Distribution of combination-wise maximum principal angles.** Number of learned generators ($m$) vs. principal angles. The blue boxes denote the interquartile range; the orange horizontal lines inside the box denote the median; the green up triangles and the blue down triangles denote the maximum and minimum of maximum principal angles, respectively; the orange dots denote the mean.

algebra dimension, so LieStoNet recovers the symmetry dimension without receiving it as input. We then compare learned and analytic generator spans using principal angles. For $m \leq 3$, we compare the learned $m$-dimensional span with each analytic $m$-generator subspace; for $m > 3$, we compare each learned 3-generator subspace with the full analytic 3-dimensional span. Figure 3 reports the resulting maximum-angle distributions. Since $L_1$ selects the correct dimension, the principal angles at that dimension mainly reflect surrogate and optimization accuracy. The smallest maximum angles occur at the selected dimension, indicating recovery of the symmetry-algebra span rather than spurious low-residual generators. Tables 4 and 5 report the selected-dimension SDE and Fokker–Planck angles.

| Ex. | Ang 1 | Ang 2 | Ang 3 |
|---|---|---|---|
| 1 | 6.261° | 2.487° | 2.313° |
| 2 | 19.370° | 11.051° | 0.571° |
| 3 | 7.643° | 1.329° | 0.276° |
| 4 | 14.292° | 6.241° | 2.505° |

*Table 4.* **Principal angles of SDE symmetry for all examples at the ground-truth dimension.**

| Ang 1 | Ang 2 | Ang 3 | Ang 4 | Ang 5 | Ang 6 |
|---|---|---|---|---|---|
| 8.417° | 7.424° | 1.829° | 1.091° | 0.752° | 0.303° |

*Table 5.* **Principal angles of Fokker-Planck equation symmetry for example 1 at the ground-truth dimension.**

In Appendix F, we additionally verify that the learned SDE generators embed into the learned Fokker–Planck generator span when both are projected to the common $(\tau, \xi)$-space.

**Higher-dimensional symmetry algebra.** We test recovery of larger algebras on the ten-dimensional Brownian SDE

$$\mathrm{d}X_t^i = \mathrm{d}W_t^i, \qquad i = 1, \ldots, 10,$$

a high-dimensional analogue of Example 1 with a 12-dimensional projectable symmetry algebra. Its generators mirror Appendix J, with translations $\partial_i$ and scaling $2t\partial_t + \sum_i x_i\partial_i$. LieStoNet recovers a 12-dimensional algebra with principal angles $0.73°, 0.79°, 0.82°, 0.93°, 0.98°, 1.14°, 1.17°, 1.38°, 1.63°, 2.08°, 10.21°, 16.55°$, supporting extension beyond the low-dimensional 3-generator benchmarks.

**Surrogate sensitivity and computational overhead.** Because LieStoNet computes symmetry losses through a learned SDE surrogate, we also evaluate sensitivity to surrogate perturbations; recovery degrades smoothly under controlled structured and random perturbations. We additionally report per-loss runtime measurements and polynomial scaling estimates for the algebraic and SDE-validity losses. These analyses are provided in Appendices H and I.

**Ablation of generator losses.** We further test whether the algebraic and SDE-validity losses are all practically useful by ablating $L_1, \ldots, L_7$ one at a time. Removing any single loss worsens the maximum principal angle between the learned and analytic symmetry-algebra spans; the largest degradations occur when removing the closure loss $L_1$ or the SDE determining-equation loss $L_6$. Removing the finite-step after-push loss $L_7$ also substantially worsens recovery, suggesting that finite-flow consistency provides useful stabilization beyond local infinitesimal invariance. The full ablation results, weight-sensitivity sweep, and qualitative generator plots are reported in Appendices O, Q, and G.

## 5. Additional Experiment Results

### 5.1. After-Push Residual (finite-$\epsilon$ symmetry check).

The defining property of an SDE symmetry is that the induced transformation maps the SDE $\mathrm{d}X_t = f(t, X_t)\mathrm{d}t + \sigma(t, X_t)\mathrm{d}W_t$ to the *same* SDE form (equivalently, it preserves the drift–diffusion coefficients under the change of variables). To test this definition beyond the infinitesimal regime, we perform a finite-$\epsilon$ "after-push" check for each learned generator $X_i = \tau_i\partial_t + \xi_i\partial_x$ (or its 2D analogue). We first sample 500 trajectories from the analytic SDE, then push every space–time point $(t, \mathbf{x})$ along the generator flow $\Phi_\epsilon = \exp(\epsilon X_i)$ to obtain $(\tilde{t}, \tilde{\mathbf{x}})$. If $X_i$ is a true symmetry direction, then the pushed trajectory should still be governed by the same coefficients, so the drift and diffusion inferred from the pushed data should match the analytic drift and diffusion evaluated at $(\tilde{t}, \tilde{\mathbf{x}})$. Operationally, we therefore fit a neural SDE surrogate on the pushed trajectories and measure the residual between its learned $(\hat{f}, \hat{\sigma})$ and the analytic $(f, \sigma)$ on a fixed evaluation box, reported separately for drift (Dft.) and diffusion (Dfu.) in Table 6. The residuals remain small across all examples and generators for $\epsilon \in \{1, 3, 5\}$, providing empirical evidence that the learned generators induce finite transformations that approximately map the SDE to itself, consistent with the definition of an SDE symmetry.

### 5.2. Real-World Stochastic Data: High-Frequency BTC/USDT

To test whether LieStoNet transfers beyond synthetic SDEs with known analytic symmetries, we apply the same pipeline without architectural or algorithmic modifications to high-frequency BTC/USDT trade data from Binance. The dataset contains approximately 1.8 million ticks over 24 hours, aggregated to 500ms resolution, filtered, and split into 575 overlapping sub-trajectories. Since no ground-truth symmetry algebra is available for this empirical system, we use the same closure-based dimension-selection criterion and finite-flow validation diagnostics used in the synthetic experiments.

| Ex. | Gen 1 | | Gen 2 | | Gen 3 | |
|---|---|---|---|---|---|---|
| | Dft. | Dfu. | Dft. | Dfu. | Dft. | Dfu. |
| 1 | 9.6 | 4.6 | 8.0 | 5.6 | 2.3 | 5.1 |
| 2 | 1.9 | 2.7 | 1.1 | 6.0 | 5.7 | 4.4 |
| 3 | 7.6 | 1.4 | 1.3 | 9.7 | 2.7 | 0.2 |
| 4 | 1.9 | 6.2 | 6.41 | 1.0 | 2.5 | 9.2 |

| Ex. | Gen 1 | | Gen 2 | | Gen 3 | |
|---|---|---|---|---|---|---|
| | Dft. | Dfu. | Dft. | Dfu. | Dft. | Dfu. |
| 1 | 8.2 | 8.9 | 9.7 | 6.4 | 8.8 | 8.5 |
| 2 | 9.1 | 0.6 | 7.3 | 8.8 | 1.0 | 8.4 |
| 3 | 7.5 | 8.1 | 9.6 | 0.9 | 6.3 | 6.2 |
| 4 | 0.8 | 8.0 | 7.9 | 8.7 | 9.5 | 8.4 |

| Ex. | Gen 1 | | Gen 2 | | Gen 3 | |
|---|---|---|---|---|---|---|
| | Dft. | Dfu. | Dft. | Dfu. | Dft. | Dfu. |
| 1 | 0.2 | 2.9 | 1.4 | 0.7 | 2.1 | 0.4 |
| 2 | 3.0 | 0.1 | 0.6 | 2.5 | 1.8 | 2.8 |
| 3 | 1.1 | 2.3 | 2.7 | 0.3 | 0.9 | 1.6 |
| 4 | 2.4 | 0.8 | 1.9 | 3.0 | 0.0 | 2.6 |

*Table 6.* After-push residuals for $\epsilon = 1, 3, 5$ from top to bottom; units are $10^{-4}$ for $\epsilon = 1$ and $\epsilon = 3$, and $10^{-3}$ for $\epsilon = 5$.

Sweeping $m = 1, 2, 3$, the first nontrivial minimum of the post-training closure loss occurs at $m^\star = 2$, with a pronounced increase from $m = 2$ to $m = 3$. This suggests a stable two-dimensional learned symmetry algebra. We then validate the two learned generators using two complementary tests. First, the after-push residual measures whether trajectories pushed along the learned generator flow remain consistent with the learned SDE drift and diffusion. Second, the distributional invariance test evaluates whether the standardized Euler–Maruyama residual distribution is preserved after pushing trajectories. Both tests indicate that the learned transformations preserve the empirical stochastic dynamics substantially better than random-push controls, as shown in Tables 7 and 8.

| $\epsilon$ | Generator 1 | | Generator 2 | |
|---|---|---|---|---|
| | Dft. | Dfu. | Dft. | Dfu. |
| 1 | $1.4 \times 10^{-3}$ | $2.3 \times 10^{-7}$ | $1.0 \times 10^{-2}$ | $2.6 \times 10^{-4}$ |
| 3 | $7.9 \times 10^{-2}$ | $9.9 \times 10^{-2}$ | $5.0 \times 10^{-2}$ | $1.7 \times 10^{-1}$ |
| 5 | $1.3 \times 10^{-1}$ | $2.7 \times 10^{-1}$ | $5.0 \times 10^{-2}$ | $2.8 \times 10^{-1}$ |

*Table 7.* **After-push residuals on BTC/USDT data.** Drift and diffusion residuals are reported after pushing trajectories along each learned generator for step length $\epsilon$. Smaller values indicate better finite-flow preservation of the learned stochastic dynamics.

| $\epsilon$ | Gen. 1 | Gen. 2 | Random control |
|---|---|---|---|
| 0.05 | 0.008 | 0.027 | 9.7 |
| 0.10 | 0.027 | 0.111 | 191.6 |
| 0.20 | 0.091 | 0.827 | 232.0 |
| 0.30 | 0.324 | 4.65 | 166.0 |
| 0.50 | 6.41 | 17.6 | 112.0 |

*Table 8.* **Distributional invariance on BTC/USDT data.** We report $\Delta\text{NLL} = \text{NLL}_{\text{pushed}} - \text{NLL}_{\text{orig}}$. The learned-generator pushes preserve the residual distribution far better than random-push controls over the tested step sizes.

## 6. Conclusion

We introduced **LieStoNet**, an end-to-end pipeline for discovering projectable Lie-point symmetries of SDEs from trajectory data. LieStoNet learns a neural SDE surrogate and then learns symmetry generators by enforcing SDE determining equations, Lie-algebraic structure, independence, and finite-flow validation losses. Across analytic SDE benchmarks, it recovers the correct symmetry dimension and strong span-level alignment with the ground-truth algebra. Ablations, after-push checks, BTC/USDT experiments, surrogate-sensitivity tests, runtime measurements, and a higher-dimensional SDE further support the proposed objectives.

**Limitations and outlook.** LieStoNet depends on the fidelity and coverage of the learned drift/diffusion surrogate, though perturbation experiments suggest smooth degradation under controlled surrogate errors. The method also restricts attention to projectable Lie-point generators. Scaling to complex high-dimensional empirical systems will require more efficient parameterizations, better surrogate calibration, and validation under partial observation and measurement noise. Future work includes extending beyond projectable symmetries and using discovered symmetries to build symmetry-constrained stochastic surrogates.

## Impact Statement

This paper presents work whose goal is to advance the field of machine learning. There are many potential societal consequences of our work, none of which we feel must be specifically highlighted here.

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

## A. Mathematical Preliminaries

### A.1. Stochastic Differential Equations

We use the Itô convention for SDEs. An $n$-dimensional Itô SDE is written as

$$\mathrm{d}\mathbf{x}_t = \mathbf{f}(t, \mathbf{x}_t)\,\mathrm{d}t + \boldsymbol{\sigma}(t, \mathbf{x}_t)\,\mathrm{d}\mathbf{W}_t, \tag{11}$$

where $\mathbf{x}_t \in \mathbb{R}^n$, $\mathbf{f} : \mathbb{R}\times\mathbb{R}^n \to \mathbb{R}^n$ is the *drift*, $\boldsymbol{\sigma} : \mathbb{R}\times\mathbb{R}^n \to \mathbb{R}^{n\times n}$ is the *diffusion*, and $\mathbf{W}_t$ is an $n$-dimensional standard Wiener process capturing the random fluctuations.

### A.2. Lie algebras: Definitions and Motivation

A *Lie algebra* is a vector space equipped with a product that captures *composition of infinitesimal transformations*. Formally, it is a real vector space $\mathfrak{g}$ whose elements $X, Y, Z \in \mathfrak{g}$ admit a bilinear operation $[\cdot, \cdot] : \mathfrak{g} \times \mathfrak{g} \to \mathfrak{g}$ (the *Lie bracket*) satisfying *antisymmetry* $[X, Y] = -[Y, X]$ and the *Jacobi identity* $[X, [Y, Z]] + [Y, [Z, X]] + [Z, [X, Y]] = 0$ along with bilinearity in each argument.

### A.3. Infinitesimal SDE Symmetry Generators as Lie Algebras

A *symmetry* of a dynamical system - deterministic or stochastic - is a transformation of variables that preserves the model's behavior. Symmetries compose: applying one symmetry after another yields another symmetry, and for smooth, continuous families this structure forms (at least locally) a *Lie group*. Many common symmetries are parameterized continuously (e.g., time shifts or state scalings), and for such families it is most convenient to work infinitesimally: what an arbitrarily small transformation does. The set of all infinitesimal symmetry directions forms a *Lie algebra*.

Concretely, consider a smooth one-parameter change of variables $(t, x) \mapsto (t', x') = \Phi_\varepsilon(t, x)$ with $\Phi_0$ equal to the identity and $\varepsilon$ being the pseudo-time component (how far is the trajectory being pushed along the generator). Its infinitesimal generator is the vector field

$$X = \tau(t, x)\,\partial_t + \xi(t, x)\,\partial_x, \tag{12}$$

(shorthanded as $(\tau, \xi)$) which acts to first order as $t \mapsto t + \varepsilon\,\tau(t, x)$ and $x \mapsto x + \varepsilon\,\xi(t, x)$. Conversely, given a vector

field $\tau, \xi$, one can recover $\Phi_\epsilon = \exp(\epsilon X)$ by integrating:

$$\frac{d}{d\varepsilon}(t(\varepsilon), x(\varepsilon)) = (\tau(t(\varepsilon), x(\varepsilon)), \xi(t(\varepsilon), x(\varepsilon)))$$

with $(t(0), x(0)) = (t, x)$.

For an Itô SDE, Itô's formula specifies how such a coordinate change transforms the drift and diffusion coefficients $(\mu, \sigma)$. We call $\Phi_\varepsilon$ a (Lie-point) symmetry if this transformation maps the SDE to the same SDE. In contrast to deterministic systems—where symmetries literally map solution trajectories to solution trajectories - SDE paths are random, so the natural invariance notion is *distributional*: a symmetry preserves the *law* of the process, meaning it maps sample paths to sample paths whose induced stochastic dynamics are equivalent in the transformed coordinates.

Finally, symmetry generators inherit an algebraic structure. They are closed under the *Lie bracket*

$$[X, Y] := XY - YX,$$

i.e. with $X = (\tau_X, \xi_X)$ and $Y = (\tau_Y, \xi_Y)$,

$$[X, Y] = (X(\tau_Y) - Y(\tau_X))\partial_t + (X(\xi_Y) - Y(\xi_X))\partial_x,$$

which captures how infinitesimal symmetries compose. This closure endows the symmetry generators with the structure of a Lie algebra. Learning multiple generators $\{X_i\}$ therefore amounts to learning a symmetry *subspace*: any linear combination is another valid infinitesimal symmetry direction, while the bracket encodes their composition structure. This Lie algebra is the chief object we aim to discover and evaluate in this work.

For instance, the two-dimensional space

$$\mathfrak{g} = \{a\,\partial_x + b\,x\partial_x : a, b \in \mathbb{R}\}$$

is closed under the Lie bracket, and a direct calculation using Lie bracket's formula gives:

$$[x\partial_x, \partial_x] \cdot f = x\partial_x\partial_x f - \partial_x(x\partial_x f) = -\partial_x f$$

So, $[x\partial_x, \partial_x] = -\partial_x$ which again lies inside $\mathfrak{g}$, showing its closure. In what follows, our symmetry generators are represented as such vector fields; the Lie bracket provides the natural notion of "closure" under composition of infinitesimal symmetries, and the resulting Lie algebra structure is the object we aim to learn.

## B. Details of Span Alignment via Principal Angles

This appendix states the mathematical procedure used to compare the *span* of learned symmetry generators to the *span* of analytic (ground-truth) generators. Because a symmetry algebra is only identifiable up to an arbitrary change

of basis (any invertible mixing of a valid generator set spans the same subspace), we evaluate agreement at the subspace level via principal angles.

**Vectorizing generators on a common domain.**  Fix an evaluation domain $\Omega \subset \mathbb{R} \times \mathbb{R}^d$ in the variables $(t, \mathbf{x})$, and choose a finite set of points $\Omega_{\text{eval}} = \{(t_p, \mathbf{x}_p)\}_{p=1}^B \subset \Omega$ where $B$ is the number of evaluation points. For a learned projectable generator

$$X_i = \tau_i(t)\,\partial_t + \xi_i(t, \mathbf{x}) \cdot \nabla_{\mathbf{x}}, \qquad i = 1, \ldots, m,$$

define its *vectorized representation* on $\Omega_{\text{eval}}$ by stacking its components:

$$v_i := \left(\tau_i(t_p), \xi_i(t_p, \mathbf{x}_p)\right)_{p=1}^B \in \mathbb{R}^{B(1+d)}. \qquad (13)$$

Collect these columns into

$$V = [v_1, \ldots, v_m] \in \mathbb{R}^{B(1+d) \times m}.$$

Similarly, given analytic generators $\{W_j\}_{j=1}^r$ with components $(\tilde{\tau}_j, \tilde{\xi}_j)$, define

$$w_j := \left(\tilde{\tau}_j(t_p), \tilde{\xi}_j(t_p, \mathbf{x}_p)\right)_{p=1}^B \in \mathbb{R}^{B(1+d)},$$

$$W = [w_1, \ldots, w_r] \in \mathbb{R}^{B(1+d) \times r}.$$

By construction, $\text{span}(V)$ and $\text{span}(W)$ represent the learned and ground-truth symmetry subspaces over $\Omega_{\text{eval}}$.

**Principal angles between subspaces.**  Let $Q_V$ and $Q_W$ be orthonormal bases for $\text{span}(V)$ and $\text{span}(W)$, e.g. from reduced QR factorizations. The principal angles $\theta_1 \leq \cdots \leq \theta_{\min(m,r)} \in [0, \pi/2]$ between the two subspaces are defined by

$$\cos(\theta_\ell) = \sigma_\ell\left(Q_V^\top Q_W\right), \quad \ell = 1, \ldots, \min(m, r),$$

where $\sigma_\ell(\cdot)$ denotes the $\ell$th singular value in descending order. Small principal angles indicate that the learned generators span (up to basis change) the same symmetry directions as the analytic generators on $\Omega_{\text{eval}}$.

**Dimension-mismatch protocol (subset comparisons).** When the learned generator count $m$ differs from the analytic dimension $r$, we compute principal angles by comparing subspaces of equal dimension:

- If $m > r$, we evaluate principal angles between $\text{span}(V_S)$ and $\text{span}(W)$ for all subsets $S \subset \{1, \ldots, m\}$ with $|S| = r$, where $V_S$ denotes the submatrix of $V$ with columns indexed by $S$.

- If $m < r$, we evaluate principal angles between $\text{span}(V)$ and $\text{span}(W_T)$ for all subsets $T \subset \{1, \ldots, r\}$ with $|T| = m$, where $W_T$ denotes the corresponding submatrix of $W$.

For each subset comparison, we report the full set of principal angles and often summarize alignment by the maximum principal angle within that subset. This yields a basis-invariant diagnostic of how closely the learned span matches the analytic symmetry span across candidate dimensions.

## C. Loss Functions

### C.1. Lie-Algebra Structure Losses

**(L1) Lie-bracket closure + constancy (fixed).** For each pair $(i,j)$, we fit coefficients $c_{ij}^k(t,x)$ such that $[X_i, X_j](t,x) \approx \sum_{k=1}^{m} c_{ij}^k(t,x) X_k(t,x)$, and penalize the projection residual:

$$L_1 := \mathbb{E}_{(t,x)\in\Omega}\left[\sum_{1\le i<j\le m}\left\|[X_i,X_j] - \sum_{k=1}^{m} c_{ij}^k X_k\right\|^2\right]$$

$$+\mathbb{E}_{(t,x)\in\Omega}\left[\sum_{1\le i<j\le m}\sum_{k=1}^{m}\|\nabla_{t,x} c_{ij}^k(t,x)\|^2\right]$$

**(L2) Jacobi identity loss.** Using the fitted coefficients $c_{ij}^k$, we penalize violations of the Jacobi identity:

$$L_2 := \sum_{\text{triples }(i,j,k)}\left\|\sum_{\text{cyclic}(i,j,k)}\sum_{\ell=1}^{m} c_{ij}^\ell\, c_{\ell k}^p\right\|^2$$

**(L3) Skew-symmetry (antisymmetry) loss.** We enforce $[X_i, X_j] = -[X_j, X_i]$:

$$L_3 := \mathbb{E}_{(t,x)\in\Omega}\left[\sum_{1\le i<j\le m}\left\|[X_i,X_j] + [X_j,X_i]\right\|^2\right]$$

**(L4) Bilinearity loss.** Sampling $a, b \sim \mathcal{U}[-1,1]$ and indices $(i,j,k)$, we penalize violations of bilinearity:

$$L_4 := \mathbb{E}\left[\left\|[aX_i + bX_j, X_k] - a[X_i,X_k] - b[X_j,X_k]\right\|^2\right]$$

$$+\mathbb{E}\left[\left\|[X_k, aX_i + bX_j] - a[X_k,X_i] - b[X_k,X_j]\right\|^2\right]$$

**(L5) Functional independence loss.** On $\Omega_{\text{span}} = \{(t_p, x_p)\}_{p=1}^{P}$, define $v_i := (\tau_i(t_p), \xi_i(t_p, x_p))_{p=1}^{P} \in \mathbb{R}^{2P}$ and $V = [v_1, \ldots, v_m] \in \mathbb{R}^{2P \times m}$, with Gram matrix $G := V^\top V$. We penalize near-singularity via

$$L_5 := -\log\det(G + \epsilon I),$$

with $\epsilon > 0$.

### C.2. SDE-Validity Losses

**(L6) SDE determining-equation loss (IIC).** We enforce the determining equations in Theorem 2.1. For each generator $X_i$ and each $(t,x) \in \Omega$, define residuals

$$r_1^{(i)}(t,x) = \xi_t + f\,\xi_x - \xi\,f_x - \partial_t(f\tau) + \tfrac{1}{2}\sigma^2\,\xi_{xx},$$

$$r_2^{(i)}(t,x) = \sigma\,\xi_x - \xi\,\sigma_x - \tau\,\sigma_t - \tfrac{1}{2}\sigma\,\tau_t,$$

where all quantities are evaluated at $(t,x)$, subscripts denote partial derivatives, and $\partial_t(f\tau) = f_t\tau + f\tau_t$. We use the squared residual loss:

$$L_6 := \frac{1}{m}\sum_{i=1}^{m}\mathbb{E}_{(t,x)\in\Omega}\left[|r_1^{(i)}(t,x)|^2 + |r_2^{(i)}(t,x)|^2\right]$$

**(L7) Prolonged pushforward residual.** The determining-equation loss is local. We add a finite-$\varepsilon$ consistency loss that checks whether the learned generator approximately pushes the *surrogate coefficients* $(f, \sigma)$ to the surrogate coefficients evaluated at the pushed point.

Fix a small $\varepsilon > 0$. For generator $X_i$, let $(t,x) \mapsto (\tilde{t}, \tilde{x})$ be the point transformation obtained by integrating

$$\frac{dt}{d\varepsilon} = \tau_i(t), \qquad \frac{dx}{d\varepsilon} = \xi_i(t,x).$$

Along this flow, we propagate coefficient fields $(\mu, \varsigma)$ using the prolonged system in our implementation:

$$\frac{d\varsigma}{d\varepsilon} = \varsigma\left(\partial_x\xi_i - \tfrac{1}{2}\partial_t\tau_i\right)$$

$$\frac{d\mu}{d\varepsilon} = \partial_t\xi_i + \mu\,\partial_x\xi_i + \tfrac{1}{2}\varsigma^2\,\partial_{xx}\xi_i - \mu\,\partial_t\tau_i,$$

initialized at $\mu(0) = f(t,x)$ and $\varsigma(0) = \sigma(t,x)$. After one $\varepsilon$-step, let $\mu_{\text{pred}}, \varsigma_{\text{pred}}$ be the propagated coefficients, and define direct evaluations at the pushed point

$$\mu_{\text{eval}} := f(\tilde{t}, \tilde{x}), \qquad \varsigma_{\text{eval}} := \sigma(\tilde{t}, \tilde{x}).$$

We penalize discrepancies and include a soft barrier discouraging negative pushed time increments along pushed trajectories:

$$L_7 := \frac{1}{m}\sum_{i=1}^{m}\mathbb{E}\left[(\mu_{\text{pred}} - \mu_{\text{eval}})^2 + (\varsigma_{\text{pred}} - \varsigma_{\text{eval}})^2\right]$$

$$+ \lambda_{\Delta t}\frac{1}{m}\sum_{i=1}^{m}\mathbb{E}\left[\text{softplus}(-\Delta\tilde{t})\right].$$

The expectations are over sampled trajectory points (coefficient terms) and adjacent time pairs (for $\Delta\tilde{t}$). We integrate the above prolonged system numerically.

## C.3. FP-Validity Losses

**(L8) Fokker-Planck Determining Equations**   This loss enforces the determining equations for Lie point symmetries of the 1D Fokker-Planck equation, given by $u_t = Au_{xx} + Bu_x + Cu$, where $A = -\frac{1}{2}\sigma^2$, $B = f - (A)_x$, and $C = f_x - (A)_{xx}$. For each generator $X_i = \tau_i(t)\partial_t + \xi_i(t,x)\partial_x + \phi_i(t,x,u)\partial_u$, with $\phi_i(t,x,u) = \beta_i(t,x)u$, the following three equations must hold:

$$R_{i,1} = (\dot{\tau}_i A + \tau_i A_t) + \xi_i A_x - 2A\xi_{i,x} \approx 0$$
$$R_{i,2} = (\dot{\tau}_i B + \tau_i B_t) - (\dot{\xi}_i + B\xi_{i,x} - \xi_i B_x) + 2A\beta_{i,x}$$
$$\qquad - A\xi_{i,xx} \approx 0$$
$$R_{i,3} = (\dot{\tau}_i C + \tau_i C_t) + \dot{\beta}_i + A\beta_{i,xx} + B\beta_{i,x} + \xi_i C_x \approx 0$$

where $\dot{(\cdot)}$ denotes $\partial_t(\cdot)$, and subscripts $t, x, xx$ denote partial derivatives. The loss function $L_8$ is the mean-squared (or mean-absolute) sum of these residuals over a batch of $(t, x)$ points:

$$L_8 := \frac{1}{m|\mathcal{B}|} \sum_{i=1}^m \sum_{(t,x)\in\mathcal{B}} (R_{i,1}^2 + R_{i,2}^2 + R_{i,3}^2)$$

**(L9) Fokker-Planck After-Flow (Pushforward on $u$)** This loss ensures that the learned generators approximately preserve the solution of the Fokker-Planck equation. For a given solution $u(t,x)$ of the FP equation, flowing $(t,x,u)$ along the prolonged generator $X_i = \tau_i\partial_t + \xi_i\partial_x + \beta_i u\partial_u$ for a small step $\varepsilon$ should result in another valid solution. Let $(t_0, x_0, u_0)$ be an initial point (where $u_0 = \hat{u}(t_0, x_0)$ is the learned FP surrogate solution). We numerically integrate the system:

$$\frac{dt}{d\alpha} = \tau_i(t)$$
$$\frac{dx}{d\alpha} = \xi_i(t,x)$$
$$\frac{du}{d\alpha} = \beta_i(t,x)u$$

from $\alpha = 0$ to $\alpha = \varepsilon$, yielding $(t_\varepsilon, x_\varepsilon, u_\varepsilon)$. The loss penalizes the difference between the flowed solution $u_\varepsilon$ and the learned FP surrogate evaluated at the flowed point, $\hat{u}(t_\varepsilon, x_\varepsilon)$:

$$L_9 := \frac{1}{m|\mathcal{B}|} \sum_{i=1}^m \sum_{(t_0,x_0)\in\mathcal{B}} (\hat{u}(t_\varepsilon, x_\varepsilon) - u_\varepsilon)^2$$

where $\mathcal{B}$ is a batch of collocation points. The integration uses Heun's method, and the residuals can be either squared (L2) or absolute (L1).

## D. Fokker–Planck Symmetries

This appendix records (i) the standard FP symmetry structure used to motivate optional FP-route losses, and (ii) proofs that the relevant sets of generators form Lie algebras.

### D.1. FP Equation Associated with an Itô SDE

For an Itô SDE in $\mathbb{R}^n$

$$\mathrm{d}x^i = f^i(t,x)\,\mathrm{d}t + \sigma^i{}_k(t,x)\,\mathrm{d}w^k,$$

the forward FP equation for a one-point density $u(x,t)$ can be written as

$$u_t + A^{ij}(t,x)u_{ij} + B^i(t,x)u_i + C(t,x)u = 0, \quad (14)$$

with coefficients (one common convention)

$$A^{ij} := -\tfrac{1}{2}(\sigma\sigma^\mathsf{T})^{ij}, \; B^i := f^i - \partial_j(\sigma\sigma^\mathsf{T})^{ij},$$
$$C := \partial_i f^i - \tfrac{1}{2}\partial_{ij}^2(\sigma\sigma^\mathsf{T})^{ij}.$$

### D.2. Projectable FP Symmetries Act Affinely in $u$

Because the FP equation is linear in $u$, any Lie-point symmetry must act affinely in $u$:

$$X = \tau(t)\partial_t + \xi^i(t,x)\partial_{x^i} + (\alpha(t,x) + \beta(t,x)u)\partial_u.$$

The "$\alpha$-part" corresponds to the linear superposition symmetry: $X_\alpha = \alpha\partial_u$ for any solution $\alpha$ of the FP equation.

### D.3. Lie Algebra Structure Proofs

**Theorem D.1** (FP symmetries form a Lie algebra). *Let* $\mathfrak{g}_{\mathrm{FP}}$ *denote the set of Lie-point symmetry generators of the FP equation. Then* $\mathfrak{g}_{\mathrm{FP}}$ *is a Lie algebra under the commutator* $[X_1, X_2] = X_1 X_2 - X_2 X_1$.

*Proof.* The infinitesimal invariance condition is linear in $X$, so $\mathfrak{g}_{\mathrm{FP}}$ is a vector space. Closure under brackets follows from the standard fact that prolongation commutes with Lie brackets: $\mathrm{pr}^{(2)}[X_1, X_2] = [\mathrm{pr}^{(2)}X_1, \mathrm{pr}^{(2)}X_2]$, together with the invariance identity $\mathrm{pr}^{(2)}X(\Delta_{\mathrm{FP}}) = \lambda_X\Delta_{\mathrm{FP}}$ for some scalar function $\lambda_X$ on jet space. Applying the commutator to $\Delta_{\mathrm{FP}}$ yields $\mathrm{pr}^{(2)}[X_1, X_2](\Delta_{\mathrm{FP}}) = \lambda_{[X_1,X_2]}\Delta_{\mathrm{FP}}$, hence $[X_1, X_2] \in \mathfrak{g}_{\mathrm{FP}}$. $\square$

## E. SDE symmetries as a Lie-subalgebra of FP Symmetries

Classical results relate SDE symmetries to a subalgebra of FP symmetries; in particular, the projectable SDE symmetry algebra $\mathfrak{g}_{\mathrm{Ito}}$ embeds into FP symmetries by adding an appropriate vertical component in $u$, and one obtains a nested structure $\mathfrak{g}_{\mathrm{Ito}} \subset \mathfrak{g}_{\mathrm{FP}}$. We now prove this containment more formally.

### E.1. Quotienting Out the Superposition Ideal and the SDE–FP Isomorphism

Let $\mathfrak{g}_{\text{Ito}}$ be the Lie algebra of *projectable* Itô symmetry generators

$$X = \tau(t)\,\partial_t + \xi^i(t,x)\,\partial_{x_i},$$

whose coefficients satisfy the Itô determining equations *(3.4)* in (Gaeta and Quintero, 1999).

Let $\mathfrak{g}_{\text{FP}}$ be the Lie algebra of projectable Lie point symmetry generators of the associated FP equation, and let

$$\mathcal{I} := \big\{ X_\alpha = \alpha(t,x)\,\partial_u \; : \; \alpha \text{ solves the FP equation} \big\}.$$

By [Thm. 3](Gaeta and Quintero, 1999), these $X_\alpha$ are precisely the "trivial" symmetries coming from linear superposition, and the general projectable FP symmetry has

$$\phi = \alpha + \beta u.$$

In particular, $\mathcal{I}$ is an (infinite-dimensional) Lie ideal in $\mathfrak{g}_{\text{FP}}$, so the quotient

$$\mathfrak{g}_{\text{FP}}^{\text{ess}} := \mathfrak{g}_{\text{FP}}/\mathcal{I}$$

is a Lie algebra.

Define the *SDE-induced essential symmetry subalgebra* as the image in the quotient of the ansatz

$$\tau\,\partial_t + \xi^i\partial_{x_i} - (\operatorname{div}\xi)\,u\,\partial_u,$$

which is exactly the "probabilistically compatible" form singled out in (Gaeta and Quintero, 1999) (see Remark 11 and equation (5.10)). Concretely,

$$\mathfrak{g}_{\text{SDE}}^{\text{ess}} := \Big\{ \big[ \tau\,\partial_t + \xi^i\partial_{x_i} - (\operatorname{div}\xi)\,u\,\partial_u \big] \in \mathfrak{g}_{\text{FP}}^{\text{ess}} \; :$$

$$(\tau,\xi) \text{ satisfy } (3.4) \Big\}.$$

**Theorem E.1** (Lie-algebra isomorphism modulo $\mathcal{I}$)**.** *Define* $\Phi : \mathfrak{g}_{\text{Ito}} \to \mathfrak{g}_{\text{FP}}^{\text{ess}}$ *as:*

$$\Phi\big(\tau\,\partial_t + \xi^i\partial_{x_i}\big) := \big[ \tau\,\partial_t + \xi^i\partial_{x_i} - (\operatorname{div}\xi)\,u\,\partial_u \big]$$

*Then $\Phi$ is a Lie-algebra isomorphism*

$$\mathfrak{g}_{\text{Ito}} \xrightarrow{\;\cong\;} \mathfrak{g}_{\text{SDE}}^{\text{ess}} \subseteq \mathfrak{g}_{\text{FP}}^{\text{ess}}.$$

**Proof**

**(1) Well-defined.** Let $X = \tau\partial_t + \xi^i\partial_{x_i} \in \mathfrak{g}_{\text{Ito}}$. By [Thm. 4](Gaeta and Quintero, 1999), $X$ extends to an FP symmetry

$$X + \phi\,\partial_u, \qquad \phi = \alpha + \beta u,$$

with $\beta = -\operatorname{div}\xi + c_0$. Imposing preservation of normalization, [Appendix B](Gaeta and Quintero, 1999) forces $c_0 = 0$ and $\beta = -\operatorname{div}\xi$. Thus

$$\tau\partial_t + \xi^i\partial_{x_i} - (\operatorname{div}\xi)\,u\partial_u$$

is an FP symmetry, and its class modulo $\mathcal{I}$ defines $\Phi(X) \in \mathfrak{g}_{\text{FP}}^{\text{ess}}$.

**(2) Linearity.** Immediate from linearity of the divergence operator and of the quotient map.

**(3) Homomorphism property.** Let

$$X_a = \tau_a\partial_t + \xi_a^i\partial_{x_i}, \qquad X_b = \tau_b\partial_t + \xi_b^i\partial_{x_i},$$

and set

$$Y_a := X_a - (\operatorname{div}\xi_a)u\partial_u, \qquad Y_b := X_b - (\operatorname{div}\xi_b)u\partial_u.$$

A direct computation of commutators on $(t,x,u)$ yields

$$[Y_a, Y_b] = [X_a, X_b] - \big(X_a(\operatorname{div}\xi_b) - X_b(\operatorname{div}\xi_a)\big)u\partial_u.$$

Writing

$$[X_a, X_b] = \tau_c\partial_t + \xi_c^i\partial_{x_i},$$

one has

$$\operatorname{div}\xi_c = X_a(\operatorname{div}\xi_b) - X_b(\operatorname{div}\xi_a),$$

and therefore

$$[Y_a, Y_b] = \tau_c\partial_t + \xi_c^i\partial_{x_i} - (\operatorname{div}\xi_c)u\partial_u.$$

Passing to classes in the quotient gives

$$[\Phi(X_a), \Phi(X_b)] = \Phi([X_a, X_b]),$$

so $\Phi$ preserves Lie brackets.

**(4) Injective.** If $\Phi(X) = 0$, then the representative

$$Y = \tau\partial_t + \xi^i\partial_{x_i} - (\operatorname{div}\xi)u\partial_u$$

lies in $\mathcal{I}$. But every element of $\mathcal{I}$ has vanishing $\partial_t$ and $\partial_{x_i}$ components, hence $\tau = \xi \equiv 0$ and $X = 0$.

**(5) Image and isomorphism.** By definition, $\mathfrak{g}_{\text{SDE}}^{\text{ess}}$ consists exactly of the classes $\Phi(X)$ with $(\tau,\xi)$ satisfying (3.4). Thus

$$\operatorname{Im}(\Phi) = \mathfrak{g}_{\text{SDE}}^{\text{ess}},$$

and $\Phi$ is a Lie-algebra isomorphism onto this subalgebra. $\square$

**Remark** (essential FP symmetries vs. superposition)

Because the FP equation is linear, its Lie point symmetry algebra contains the infinite-dimensional family of solution-translation symmetries

$$X_\alpha = \alpha(t,x)\partial_u,$$

where $\alpha$ is any solution of the FP equation. In [Thm. 3](Gaeta and Quintero, 1999) these appear as the $\alpha$-term in $\phi = \alpha + \beta u$ and are explicitly identified as the symmetries implied by linear superposition. Quotienting by the ideal

$$\mathcal{I} = \{X_\alpha\}$$

removes precisely these trivial directions and yields the *essential* FP symmetry algebra $\mathfrak{g}_{FP}^{ess} = \mathfrak{g}_{FP}/\mathcal{I}$. In this quotient, the map $\Phi$ identifies each genuine Itô symmetry $(\tau, \xi)$ with the unique normalization-compatible FP symmetry class whose $u$-component is fixed by $\beta = -\operatorname{div} \xi$ (cf. (Gaeta and Quintero, 1999), Remark 11, equation (5.10), and Appendix B).

## F. SDE–Fokker–Planck Subalgebra Validation

The theoretical relation in Appendix E implies that projectable SDE symmetry generators embed into the Fokker–Planck symmetry algebra. Thus, learned SDE generators should form a subalgebra of the learned Fokker–Planck generators when both are projected to the common $(\tau, \xi)$-space, discarding the $u$-component of the Fokker–Planck generators.

To test this relation empirically, we train both SDE and Fokker–Planck generators for Example 1 and compare the span of the learned SDE generators with the projected $(\tau, \xi)$-span of the learned Fokker–Planck generators using principal angles. The resulting angles are $2.00°$, $13.63°$, $18.81°$. These small angles confirm that the learned SDE generators are recovered as a subalgebra of the learned Fokker–Planck generators, providing a direct quantitative validation of the SDE–Fokker–Planck connection beyond qualitative comparison.

## G. Plots and Heat Maps of Learned SDE Symmetry Generators

In this section, we present plots and heatmaps of the learned symmetry generators. As discussed in the main text and in the principal-angle evaluation methodology, the discovered functions are generally close to linear combinations of the ground-truth symmetries associated with each SDE. Consequently, they may not admit direct interpretation in terms of functional form or scale relative to the underlying ground-truth symmetry functions. These visualizations are included primarily for completeness and to provide qualitative insight into the types of functions ultimately learned by LieStoNet.

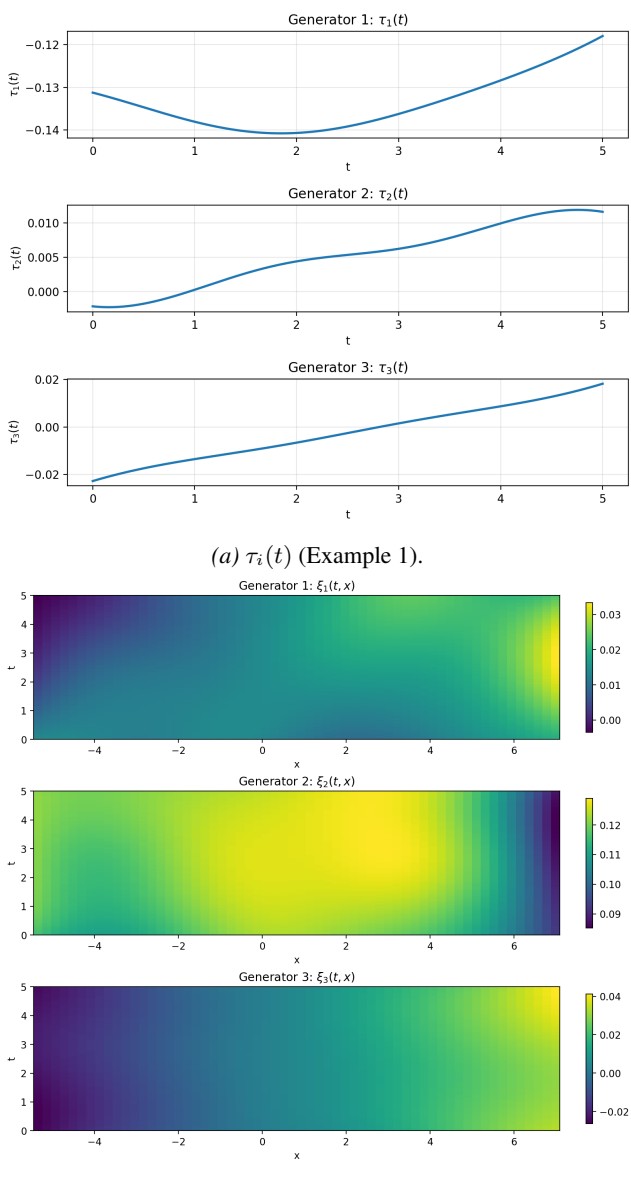

*(a)* $\tau_i(t)$ *(Example 1).*

*(b)* $\xi_i(t, x)$ *(Example 1).*

*Figure 4.* Learned generator components for Example 1. Part (a): temporal component $\tau_i(t)$. Part (b): spatial component $\xi_i(t, x)$ visualized as heat maps.

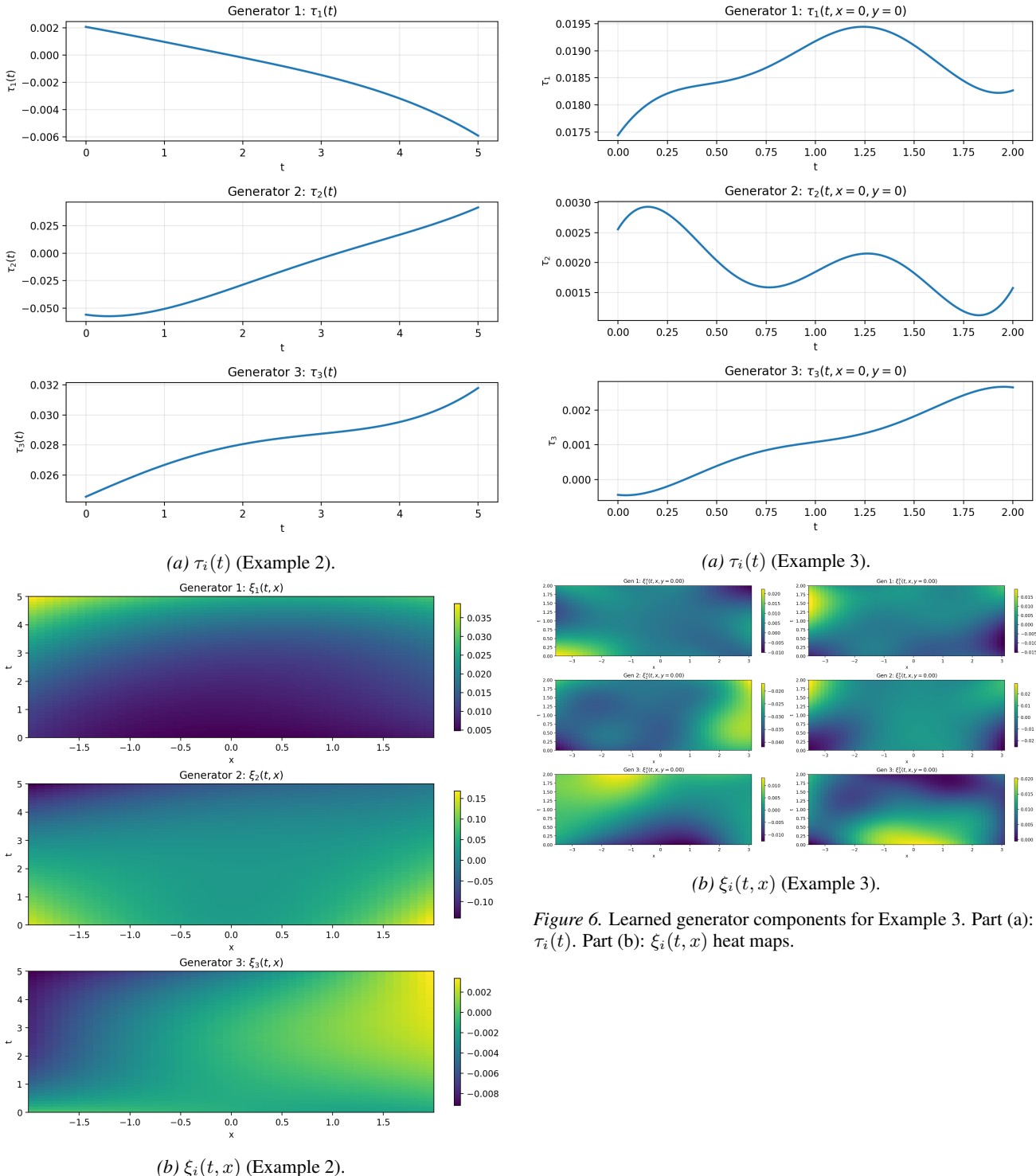

*(a)* $\tau_i(t)$ (Example 2).

*(b)* $\xi_i(t, x)$ (Example 2).

*Figure 5.* Learned generator components for Example 2. Part (a): $\tau_i(t)$. Part (b): $\xi_i(t, x)$ heat maps.

*(a)* $\tau_i(t)$ (Example 3).

*(b)* $\xi_i(t, x)$ (Example 3).

*Figure 6.* Learned generator components for Example 3. Part (a): $\tau_i(t)$. Part (b): $\xi_i(t, x)$ heat maps.

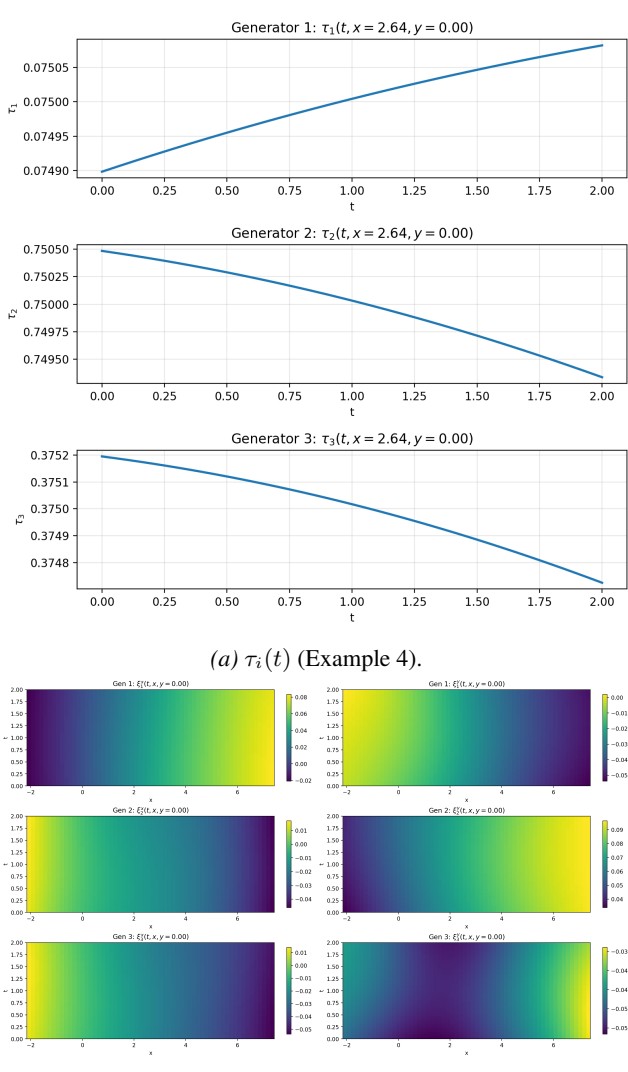

*(a) $\tau_i(t)$ (Example 4).*

*(b) $\xi_i(t, x)$ (Example 4).*

*Figure 7.* Learned generator components for Example 4. Part (a): $\tau_i(t)$. Part (b): $\xi_i(t, x)$ heat maps.

## H. Sensitivity to Surrogate Approximation Error

LieStoNet computes the symmetry losses using the learned neural SDE surrogate $(\hat{f}, \hat{\sigma})$, so surrogate misspecification can affect the recovered generators. Let $(f, \sigma)$ denote the true drift and diffusion, $G^\star$ a true symmetry generator, $\hat{G}$ a learned generator, and $D_{f,\sigma}(\cdot)$ the SDE determining-equation operator. A standard perturbation argument gives

$$\|D_{f,\sigma}(\hat{G})\| \leq \|D_{\hat{f},\hat{\sigma}}(\hat{G})\| + C(\hat{G})\Big(\|\hat{f} - f\|_{C^1} + \|\hat{\sigma} - \sigma\|_{C^1}\Big),$$

where $C(\hat{G})$ depends on the size and smoothness of the learned generator. Thus, the determining-equation residual under the true SDE is controlled by two terms: the residual minimized during training under the surrogate, and the

$C^1$ approximation error of the surrogate itself. Under a local stability condition for the determining operator, this implies that symmetry recovery degrades continuously with surrogate error (for $C^\star$ being the local stability constant):

$$\inf_{A \in GL(k)} \|\hat{G} - AG^\star\| \leq C^\star\Big(\|D_{\hat{f},\hat{\sigma}}(\hat{G})\| +$$

$$\|\hat{f} - f\|_{C^1} + \|\hat{\sigma} - \sigma\|_{C^1}\Big).$$

Empirically, we perturb the surrogate in Example 1 using structured sine perturbations and random MLP perturbations. The maximum principal angle and determining-equation loss increase smoothly with perturbation amplitude, rather than exhibiting abrupt instability.

| $\delta$ | Sine perturbation | | Random MLP perturbation | |
|---|---|---|---|---|
| | $\theta$ | $L_6$ | $\theta$ | $L_6$ |
| 0 | $1.8°$ | $3.7 \times 10^{-2}$ | $2.1°$ | $4.2 \times 10^{-2}$ |
| 0.02 | $3.1°$ | $5.8 \times 10^{-2}$ | $2.2°$ | $4.5 \times 10^{-2}$ |
| 0.1 | $12.2°$ | $9.6 \times 10^{-2}$ | $2.8°$ | $1.1 \times 10^{-1}$ |
| 0.5 | $35.9°$ | $1.2 \times 10^{-1}$ | $14.8°$ | $2.1 \times 10^{-1}$ |

*Table 9.* **Sensitivity to surrogate perturbations.** Here $\delta$ is perturbation amplitude, $\theta$ is the maximum principal angle, and $L_6$ is the SDE determining-equation loss. Recovery degrades smoothly as the surrogate is perturbed.

## I. Computational Overhead of Loss Terms

The main computational cost in Stage 2 comes from repeated automatic differentiation of the generator networks, especially for losses involving Jacobians and Hessians. Let $D_g^{(1)}$ and $D_g^{(2)}$ denote the cost of evaluating first- and second-order derivatives of one generator network, $B$ the number of sampled space–time points in a minibatch, $m$ the number of learned generators, $n$ the state dimension, and $N$ the number of subsampled trajectory time points. The algebraic and SDE-validity losses scale polynomially in these quantities:

$$L_1, L_3 = O\Big(B(mD_g^{(1)} + m^2)\Big),$$
$$L_2 = O\Big(B(mD_g^{(2)} + m^3)\Big),$$
$$L_4 = O\Big(B(mD_g^{(1)} + m^3)\Big),$$
$$L_6 = O(BmD_g^{(2)}),$$
$$L_7 = O(2nNmD_g^{(2)}).$$

Thus the cost grows polynomially rather than combinatorially in the number of generators.

We also measure per-loss wall-clock time for one representative setting, $m = 3$, $B = 2048$, on an NVIDIA H100 GPU.

The finite-step after-push loss $L_7$ is the most expensive term, while the algebraic losses together account for roughly half of the measured total. This overhead is empirically useful: the ablation in Appendix O shows that removing the algebraic losses substantially worsens symmetry recovery.

| Loss | $L_1$ | $L_2$ | $L_3$ | $L_4$ | $L_5$ | $L_6$ | $L_7$ |
|------|-------|-------|-------|-------|-------|-------|-------|
| Time (ms) | 0.43 | 0.76 | 0.38 | 0.52 | 0.47 | 0.47 | 2.15 |

*Table 10.* **Per-loss wall-clock time.** Measured for $m = 3$, $B = 2048$, on an NVIDIA H100 GPU.

## J. Complete Lie Point Symmetry Derivation for Example 1: $dx = \sigma_0 dW(t)$

We recall from Theorem 2.1 that a symmetry generator for an SDE must satisfy the determining equations

$$\xi_t + f\xi_x - \xi f_x - \partial_t(f\tau) + \tfrac{1}{2}\sigma^2\xi_{xx} = 0,$$

$$\sigma\xi_x - \xi\sigma_x - \tau\sigma_t - \tfrac{1}{2}\sigma\tau_t = 0.$$

With $f = 0, \sigma = \sigma_0$ the determining equations become $\xi_t + \tfrac{1}{2}\sigma_0^2\xi_{xx} = 0$ and $\xi_x = \tfrac{1}{2}\tau_t$. Since $\tau$ is only a function of $t$, the second one implies that $\xi$ is linear in $x$, substituting which into the first one gives $\xi_t = 0$. Thus, $\xi(t,x) = c_1x + c_2$. Substituting this in $\xi_x = \tfrac{1}{2}\tau_t$ and solving gives $\tau(t) = 2c_1t + c_3$. Thus, a general symmetry has the form

$$X_0 = \tau(t)\partial_t + \xi(x)\partial_x = (2c_1t + c_3)\partial_t + (c_1x + c_2)\partial_x.$$

We find that there are three independent constants of integration implying that the symmetry Lie algebra is three dimensional. Choosing one constant as non-zero while others being zero gives a simple basis for these generators producing:

$$c_3 = 1 : v_1 = \partial_t$$
$$c_2 = 1 : v_2 = \partial_x$$
$$c_1 = 1 : v_3 = 2t\partial_t + x\partial_x$$

## K. Complete Lie Point symmetry derivation for example 2: $dx = xdt + dW(t)$

In this case, with $f = x, \sigma = 1$, the second determining equation gives $\xi_x = \tfrac{1}{2}\tau_t$. Differentiating w.r.t. $x$ and then integrating gives $\xi(t,x) = a(t)x + b(t)$ implying $\tau_t = 2a(t)$. We now plug these into the first equation resulting in

$$a'(t)x + \big(b'(t) - b(t)\big) - x\tau_t = 0$$

Since this is a linear expression in $x$ that's identically 0, the coefficients must be 0 separately, producing $a' = \tau_t, b' = b$. Combined with $\tau_t = 2a(t)$ this gives $a(t) = c_1e^{2t}, b(t) =$

$c_2e^t$. Substituting these into our previous expressions for $\xi, \tau$ we get that a general symmetry has the form:

$$X = \big(c_1e^{2t} + c_3\big)\partial_t + \big(c_1e^{2t}x + c_2e^t\big)\partial_x$$

Similarly as before, choosing values for the constants with one non-zero constant at a time gives the following three generators as a basis:

$$c_3 = 1 : v_1 = \partial_t$$
$$c_2 = 1 : v_2 = e^t\partial_x$$
$$c_1 = 1 : v_3 = e^{2t}(\partial_t + x\partial_x)$$

## L. Complete Lie Point Symmetry Derivation for Example 3: $dx = ydt$, $dy = -k^2ydt + \sqrt{2k^2}dW(t)$

This is a two dimensional example with drift and diffusion given by:

$$f = \begin{pmatrix} y \\ -k^2y \end{pmatrix}, \quad \sigma = \begin{pmatrix} 0 & 0 \\ 0 & \sqrt{2k^2} \end{pmatrix}$$

The general symmetry looks like

$$X_0 = \tau(t)\partial_t + \xi^1(t,x,y)\partial_x + \xi^2(t,x,y)\partial_y$$

with the determining equations as

$$\partial_t\xi^i + f^j\partial_j\xi^i - \xi^j\partial_j f^i - \partial_t(f^i\tau) + \tfrac{1}{2}(\sigma\sigma^T)^{jk}\partial_{jk}^2\xi^i = 0,$$

$$(\sigma^j{}_k\partial_j)\xi^i - (\xi^j\partial_j)\sigma^i{}_k - \tau\partial_t\sigma^i{}_k - \tfrac{1}{2}\sigma^i{}_k\tau_t = 0.$$

We first use the second equation to constrain $\xi^1, \xi^2$. Since $\sigma_2^2$ is the only non-zero component, the non-trivial constraints come from $k = 2$.

- For $(i = 1, k = 2)$: $(\sigma^j{}_2\partial_j\xi^1 = 0 \Rightarrow \partial_y\xi^1 = 0)$. Thus, $\xi^1 = a(x, t)$ (no $y$-dependence).

- For $(i = 2, k = 2)$: $(\sigma^j{}_2\partial_j\xi^2 - \tfrac{1}{2}\sigma^2{}_2\tau_t = 0 \Rightarrow \partial_y\xi^2 = \tfrac{1}{2}\tau_t)$. Hence $\xi^2 = \tfrac{1}{2}\tau_t y + g(x, t)$.

We now use the first determining equation. Note that $\sigma\sigma^T = \text{diag}(0, 2k^2)$, so the diffusion term is $\tfrac{1}{2}(\sigma\sigma^T)^{jk}\partial_{jk}^2\xi^i = k^2\partial_{yy}^2\xi^i$; but $\xi^1$ is independent of $y$, and $\xi^2$ is at most linear in $y$, so $\partial_{yy}^2\xi^i = 0$ and the diffusion term drops out for the first equation for both $i = 1, 2$. The first determining equation then gives

- For $i = 1$, with $f^1 = y$, $f^2 = -k^2y$, one gets $a_t + ya_x - \xi^2 - y\tau_t = 0$. Substitute $\xi^2 = \tfrac{1}{2}\tau_t y + g(x, t)$ to get $a_t - g + y\big(a_x - \tfrac{3}{2}\tau_t\big) = 0$. Thus

$$a_x = \tfrac{3}{2}\tau_t, \quad g = a_t.$$

- For $i = 2$ the first determining equation on substituting the drift and diffusion gives

$$\left(\tfrac{1}{2}\tau_{tt}y + g_t\right) + \left(yg_x - \tfrac{1}{2}k^2\tau_t y\right)$$

$$+k^2\left(\tfrac{1}{2}\tau_t y + g\right) + k^2 y\tau_t = 0$$

which on simplifying is

$$y\left(\tfrac{1}{2}\tau_{tt} + g_x + k^2\tau_t\right) + \left(g_t + k^2 g\right) = 0.$$

Since a linear expression in $y$ being identically 0 implies that the coefficients are each 0, we get $g_t + k^2 g = 0$ and $\tfrac{1}{2}\tau_{tt} + g_x + k^2\tau_t = 0$. In the second equation, we use the fact that $g_x = a_{tx} = \tfrac{3}{2}\tau_{tt}$, giving us $2\tau_{tt} + k^2\tau_t = 0$.

Integrating $a_x = \tfrac{3}{2}\tau_t$ with respect to $x$ gives $a(x,t) = \tfrac{3}{2}\tau_t x + h(t)$ for some $h$. Then $g = a_t = \tfrac{3}{2}\tau_{tt}x + h'(t)$. Substituting this in $g_t + k^2 g = 0$ gives

$$\frac{3}{2}x(\tau_{ttt} + k^2\tau_{tt}) + h'' + k^2 h = 0.$$

The coefficient of $x$ must be 0 for this expression to be identically 0, thus $\tau_{ttt} + k^2\tau_{tt} = 0$. Along with the previously found relation $2\tau_{tt} + k^2\tau_t = 0$, this yields $\tau_{tt} = 0$ thus implying $\tau_t = 0$ i.e. $\tau(t) = c_1$. Using $\tau_t = c_1$ in $a_x = \tfrac{3}{2}\tau_t$ implies $a(x,t) = a(t)$ is independent of $x$. Thus, $g(x,t) = a_t$ is also independent of $x$. Solving $g_t + k^g = 0$ yields $g(t) = c_2 r^{-k^2 t}$. Substituting this and integrating $g = a_t$ gives $a(t) = c_3 - \tfrac{c_2}{k^2}e^{-k^2 t}$. Thus, a general symmetry looks like:

$$X = c_1\partial_t + \left(c_3 - \frac{c_2}{k^2}e^{-k^2 t}\right)\partial_x + c_2 e^{-k^2 t}\partial_y.$$

Choosing one constant non-zero at a time while others as zero gives the following set as generators:

$$c_1 = 1 : v_1 = \partial_t$$
$$c_3 = 1 : v_2 = \partial_x$$
$$c_2 = 1 : v_3 = e^{-k^2 t}(k^{-2}\partial_x - \partial_y)$$

## M. Complete Lie Point Symmetry Derivation for Example 4: $dx = (a_1/x)dt + dW_1$; $dy = a_2 dt + dW_2$

$$f = \begin{pmatrix} a_1/x \\ a_2 \end{pmatrix}, \quad \sigma = \begin{pmatrix} 1 & 0 \\ 0 & 1 \end{pmatrix}$$

Once again, the general symmetry looks like

$$X_0 = \tau(t)\partial_t + \xi^1(t,x,y)\partial_x + \xi^2(t,x,y)\partial_y$$

with the determining equations as

$$\partial_t\xi^i + f^j\partial_j\xi^i - \xi^j\partial_j f^i - \partial_t(f^i\tau) + \tfrac{1}{2}(\sigma\sigma^T)^{jk}\partial_{jk}^2\xi^i = 0,$$

$$(\sigma^j{}_k\partial_j)\xi^i - (\xi^j\partial_j)\sigma^i{}_k - \tau\partial_t\sigma^i{}_k - \tfrac{1}{2}\sigma^i{}_k,\tau_t = 0.$$

$\sigma$ is constant so that $(\xi\cdot\nabla)\sigma = 0 = \partial_t\sigma$. Again, we use the second equation to get the form of $\xi$, which gives us $\partial_k\xi^i - \tfrac{1}{2}\delta^i_k\tau_t = 0$. For different $i, k$ we have:

$$i = 1, k = 1 : \xi^1_x = \frac{1}{2}\tau_t$$

$$i = 1, k = 2 : \xi^1_y = 0$$

$$i = 2, k = 1 : \xi^2_x = 0$$

$$i = 2, k = 2 : \xi^2_y = \frac{1}{2}\tau_t$$

Integrating these gives

$$\xi^1(t,x,y) = \frac{1}{2}\tau_t(t)x + g_1(t)$$

$$\xi^2(t,x,y) = \frac{1}{2}\tau_t(t)y + g_2(t)$$

Thus, $\xi$ are at most linear in $x, y$ and second spatial derivatives all vanish. The first determining equation for $i = 1$ on plugging the expressions for $\xi^1, \xi^2$, along with the drift and diffusion functions gives the following.

- For $i = 1$ this gives

$$\tfrac{1}{2}\tau_{tt}x + g'_1(t) + \frac{a_1 g_1(t)}{x^2} = 0.$$

For this equality to hold, the coefficients of various powers of $x$ should separately be 0. Thus we get $\tau_{tt} = 0 = g_1(t)$. Hence,

$$\tau(t) = c_1 t + c_2, \quad \xi^1(x,y,t) = \frac{c_1}{2}x.$$

- For $i = 2$ the determining equation gives

$$\frac{1}{2}\tau_{tt}y + g'_2 - \frac{a_2}{2}\tau_t = 0.$$

Equating coefficients in $y$ gives $\tau_{tt} = 0$ (consistent with earlier) and $g'_2 = \frac{a_2}{2}\tau_t = \frac{a_2}{2}c_1$. On integrating the last equation we get $g_2(t) = \frac{a_2}{2}c_1 t + c_3$.

The general symmetry thus looks like

$$X = (c_1 t + c_2)\partial_t + \frac{c_1}{2}x\partial_x + \left(\frac{c_1}{2}(y + a_2 t) + c_3\right)\partial_y.$$

Choosing one constant as non-zero at a time gives the following basis of generators:

$$c_2 = 1 : v_1 = \partial_t$$
$$c_3 = 1 : v_2 = \partial_y$$
$$c_1 = 1 : v_3 = 2t\partial_t + x\partial_x + (y + a_2 t)\partial_y$$

## N. Complete Derivation for Fokker-Planck Symmetries for Example 1: $dx = \sigma_0 dw(t)$

For the 1-d brownian motion $dx = \sigma_0 dw(t)$, the associated FP equation is $u_t = \frac{\sigma_0^2}{2} u_{xx}$. Following the notation from (5) , this means $A = -\frac{\sigma_0^2}{2}, B = 0, C = 0$. The three FP determining equations from theorem 2.2 boil down to:

$$\xi_x = \frac{1}{2}\tau_t, \quad -\xi_t + 2A\beta_x - A\xi_{xx} = 0, \quad \beta_t + A\beta_{xx} = 0.$$

The first equation easily solves to $\xi(t,x) = \frac{1}{2}\tau_t(t)x + k(t)$ where $k(t)$ is any function of $t$. Substituting this result into the second equation gives

$$\beta_x = \frac{1}{2A}\left(\frac{1}{2}\tau_{tt}x + k'\right) = \frac{\tau_{tt}}{4A}x + \frac{k'}{2A}$$

which upon integrating with respect to $x$ gives:

$$\beta(t,x) = \frac{\tau_{tt}}{8A}x^2 + \frac{k'}{2A}x + m(t)$$

for some $m(t)$. Substituting this into the third equation gives

$$\beta_t + A\beta_{xx} = \left(\frac{\tau_{ttt}}{8A}x^2 + \frac{k''}{2A}x + m'(t)\right) + A \cdot \frac{\tau_{tt}}{4A} = 0$$

This is a quadratic in $x$ which is identically 0. Thus, each coefficient in the quadratic must vanish separately, giving us

$$\tau_{ttt} = 0, \quad k'' = 0, \quad m'(t) + \frac{1}{4}\tau_{tt} = 0.$$

Integrating these gives:

$$\tau(t) = at^2 + bt + c, \quad k(t) = dt + e, \quad m(t) = -\frac{a}{2}t + f.$$

Substituting these into expressions for $\beta$ and $\xi$ from earlier, we get:

$$\tau = at^2 + bt + c$$

$$\xi = \tfrac{1}{2}\tau_t x + dt + e$$

$$\beta = \frac{\tau_{tt}}{8A}x^2 + \frac{d}{2A}x - \frac{a}{2}t + f.$$

Thus, there are six independent parameters/constants $a, b, c, d, e, f$ resulting in a six dimensional vector space for the set of symmetries. We may choose any 6 linearly independent vectors for $(a, b, c, d, e, f)$ to get a vector-space basis for the Lie algebra. We choose the 6 vectors where each of the parameters except for one parameter is zero. Using these choices in the general expression for a symmetry, $V = \tau\partial_t + \xi\partial_x + \beta u\partial_u$, gives us the following six familiar set of symmetry generators:

$$c = 1 : v_1 = \partial_t$$

$$e = 1 : v_2 = \partial_x$$

$$f = 1 : v_3 = u\partial_u$$

$$d = 1 : v_4 = t\partial_x - \frac{x}{\sigma_0^2}u\partial_u$$

$$b = 1 : v_5 = t\partial_t + \frac{x}{2}\partial_x$$

$$a = 1 : v_6 = t^2\partial_t + tx\partial_x - \frac{1}{2}\left(t + \frac{x^2}{\sigma_0^2}\right)u\partial_u$$

## O. Ablation Study for Generator-Training Losses

We ablate the generator-training losses $L_1, \ldots, L_7$ by removing one loss at a time while keeping all other experimental settings fixed. The results are shown in table 12. In both examples, removing any single loss term worsens recovery, as measured by the maximum principal angle between the learned and analytic symmetry-algebra spans. This supports the practical role of each loss in addition to its theoretical motivation.

The degradation is consistent with the intended function of the losses. Removing $L_1$, which enforces Lie-bracket closure, or $L_6$, which enforces the SDE determining equations, causes the largest deterioration. Removing $L_2$ and $L_3$, corresponding to algebraic identities, has a milder effect in Example 2 but a stronger effect in Example 1. Removing $L_7$, the finite-step after-push loss, also degrades performance, indicating that the local infinitesimal constraints in $L_6$ alone are not sufficient for robust recovery in practice.

## P. Implementation Details for Learning SDE Symmetries

This section documents the exact experimental/implementation choices used in the accompanying notebooks for *SDE symmetry learning* (all Fokker–Planck symmetry components are intentionally omitted). Across all examples, the pipeline has two stages:

1. **Neural SDE surrogate (Stage 1).** Fit a neural surrogate for the drift and diffusion from increment data using an increment-likelihood objective.

2. **Generator learning (Stage 2).** Fit neural symmetry generators $(\tau, \xi)$ (and an auxiliary $\beta$ head for code-compatibility that is *unused* for the SDE-only write-up) by minimizing a weighted sum of algebraic consistency terms (closure/Jacobi/etc.) and the Ito determining equations / pushforward constraints computed using the learned surrogate coefficients.

| Example | State | $(T, \Delta t)$ | $N$ | $n_{\text{traj}}$ (Stage 1) | $B = n_{\text{traj}}N$ | Stage-1 surrogate |
|---|---|---|---|---|---|---|
| 1 (Brownian) | 1D | (5.0, 0.01) | 500 | 2048 | 1,024,000 | MLP $[2, 64, 64, 2]$, `tanh`, $\hat{\sigma} =$ softplus $+ 10^{-3}$ |
| 2 ($dx = x\, dt + \sigma\, dW$) | 1D | (5.0, 0.01) | 500 | 512 | 256,000 | MLP $[2, 64, 64, 2]$, `tanh`, i.i.d. increment dataset over $\|x\| \le 2$ |
| 3 (2D linear) | 2D | (2.0, 0.01) | 200 | 2048 | 409,600 | MLP $[3, 64, 64, 4]$, `tanh`, diagonal $\hat{\sigma} \in \mathbb{R}^2$ |
| 4 (2D, PSD diffusion) | 2D | (2.0, 0.01) | 200 | 2048 | 409,600 | Drift MLP + Cholesky MLP, `swish`, $\Sigma = LL^{\top}$ |

*Table 11.* Stage-1 (surrogate) dataset sizes and architectures. Here $N = \lfloor T/\Delta t \rfloor$ and $B$ counts increment samples (one per time step per trajectory).

| | Full | w/o $L_1$ | w/o $L_2$ | w/o $L_3$ | w/o $L_4$ | w/o $L_5$ | w/o $L_6$ | w/o $L_7$ |
|---|---|---|---|---|---|---|---|---|
| Example 1 | 6.26° | 55.56° | 25.89° | 35.05° | 36.60° | 39.77° | 86.78° | 44.54° |
| Example 2 | 19.37° | 72.37° | 25.81° | 26.56° | 37.86° | 42.62° | 86.39° | 57.00° |

*Table 12.* **Ablation of generator-training losses.** Entries are maximum principal angles between learned and analytic symmetry-algebra spans; smaller is better. "Full" uses all losses $L_1, \ldots, L_7$, while each ablated column removes exactly one loss.

All notebooks run in `jax` with `jax_enable_x64=True` and use float64 throughout for stability.

### P.0.1. STAGE 1: NEURAL SDE SURROGATE TRAINING

**Increment dataset construction.** Let $t_n = n\Delta t$ with $\Delta t = \text{dt}$ and $n = 0, \ldots, N-1$, where $N = \lfloor T/\Delta t \rfloor$. The Stage-1 dataset consists of pairs $(z_n, \Delta X_n)$ where $\Delta X_n = X_{n+1} - X_n$ and

$$z_n = \begin{cases} (t_n, x_n) & \text{(1D examples)} \\ (t_n, x_n, y_n) & \text{(2D time-dependent example)} \\ (x_n, y_n) & \text{(2D time-homogeneous example)} \end{cases}$$

For simulated path data (Examples 1, 3, 4), the notebook forms all increments from the full trajectory bank, yielding $B = n_{\text{traj}} \cdot N$ training samples. Example 2 is special: Stage 1 is built as an *i.i.d. increment dataset* by sampling $(t, x)$ values and drawing $\Delta x$ directly from the Euler increment distribution for the target SDE; a pseudo-"trajectory" tensor is only created for downstream API compatibility.

**Input normalization.** In all Stage-1 surrogates, inputs are z-scored:

$$\tilde{z} = (z - \mu_z)/(\sigma_z + 10^{-8}),$$

where $(\mu_z, \sigma_z)$ are empirical mean/std over the full Stage-1 input cloud.

**Surrogate parameterizations and likelihood.**

- **Examples 1 & 2 (1D).** A single MLP maps $\tilde{z} = (\tilde{t}, \tilde{x})$ to two outputs $(\hat{f}, g)$, with diffusion enforced positive via
  $$\hat{\sigma}(\tilde{z}) = \text{softplus}(g(\tilde{z})) + \sigma_{\min}.$$

The increment model is $\Delta x \sim \mathcal{N}(\hat{f}\Delta t, \ \hat{\sigma}^2\Delta t)$ and the loss is the per-sample Gaussian negative log-likelihood (dropping constants) averaged over mini-batches.

- **Example 3 (2D, diagonal diffusion).** A single MLP maps $\tilde{z} = (\tilde{t}, \tilde{x}, \tilde{y})$ to $(\hat{f}_x, \hat{f}_y, g_x, g_y)$ and sets $\hat{\sigma}_i = \text{softplus}(g_i) + \sigma_{\min}$. The likelihood is a factorized diagonal Gaussian for $\Delta(x, y)$.

- **Example 4 (2D, PSD diffusion via Cholesky).** Two MLPs are trained: a drift network $\hat{f}(x, y)$ and a diffusion-Cholesky network returning $(\ell_{11}^{\text{raw}}, \ell_{21}, \ell_{22}^{\text{raw}})$, with

$$\ell_{11} = \text{softplus}(\ell_{11}^{\text{raw}}) + \sigma_{\text{floor}}$$
$$\ell_{22} = \text{softplus}(\ell_{22}^{\text{raw}}) + \sigma_{\text{floor}},$$
$$L = \begin{pmatrix} \ell_{11} & 0 \\ \ell_{21} & \ell_{22} \end{pmatrix}; \Sigma = LL^{\top}.$$

The increment model is multivariate Gaussian $\Delta X \sim \mathcal{N}(\hat{f}\Delta t, \ \Sigma\Delta t)$.

**Regularization and optimizers.**

- **Examples 1/2/3:** The Stage-1 objective adds explicit L2 weight decay as `weight_decay *` `l2_tree(params)` (even though the optimizer is plain Adam). Optimization uses `optax.adam(lr)` with minibatches sampled *without replacement* from the full increment dataset each step.

- **Example 4:** Drift and diffusion-Cholesky networks are optimized with separate `adamw` optimizers (weight_decay= 0) and global-norm gradient clipping. Training alternates multiple drift steps and fewer

diffusion steps per outer iteration. Additional diffusion regularizers are used: (i) a JVP-based smoothness penalty on $L(x, y)$, (ii) a minibatch variance stabilizer on $\text{diag}(L)$, and (iii) a whitening/calibration penalty encouraging $z = L_{\Delta t}^{-1}(\Delta X - \hat{f}\Delta t) \sim \mathcal{N}(0, I)$.

### P.0.2. STAGE 2: SYMMETRY GENERATOR PARAMETERIZATION AND TRAINING

**Training point cloud for Stage 2.** All generator losses are evaluated on minibatches of a point cloud in the extended space $(t, x)$ or $(t, x, y)$:

$$\mathcal{T} = \{(t_n, x_n)\} \text{ or } \{(t_n, x_n, y_n)\}.$$

The source of this cloud differs by example:

- **Examples 1 & 2:** simulate the learned surrogate by Euler–Maruyama using `CFGGen` with $n_{\text{traj}} = 256$, and flatten all $(N+1)$ states, giving $B_{\text{gen}} = n_{\text{traj}}(N+1)$ points.

- **Example 3:** simulate the learned surrogate with $n_{\text{traj}} = 256$ but additionally (i) enforce domain control in $x \in [x_{\min}, x_{\max}]$ via clipping at evaluation time and reflection after stepping, (ii) optionally *oversample* candidate trajectories (`traj_balance_oversample=12`) and greedily select a subset whose histogram occupancy over $x$-bins is closer to uniform, and (iii) build $\mathcal{T}$ as a fixed-size *mixture* of trajectory points and uniform box samples with fraction `tx_mix_uniform_frac=0.7` (total point count kept constant).

- **Example 4:** constructs `TX_gen` directly from the available $(t, XY)$ tensor by flattening all observed path states, yielding $B_{\text{gen}} = n_{\text{traj}}(N+1)$ points with the full data trajectory bank. Its minibatch sampler mixes half empirical points and half uniform box samples in bounds inferred from `TX_gen`, with an $x$-floor to avoid instability near $x = 0$.

**Generator neural architectures.** Each generator $i = 1, \ldots, m$ is represented by neural fields:

$$\tau_i(t) \in \mathbb{R}, \ \xi_i(t, x) \in \mathbb{R} \text{ (1D) or } \xi_i(t, x, y) \in \mathbb{R}^2 \text{ (2D)}.$$

In code, each generator uses separate MLPs for $\tau$ and $\xi$. A third head $\beta$ is also instantiated for compatibility with shared utilities but is *not used* in the SDE-only description here.

- **Examples 1 & 2:** `tanh` MLPs with fixed widths: $\tau$: $[1, 32, 32, 1]$; $\xi$: $[2, 64, 64, 1]$ in 1D (Ex. 1) and $[3, 64, 64, 2]$ in 2D (Ex. 4). $m = 3$ (Ex. 1) and $m = 3$ (Ex. 4).

- **Example 3:** deeper `swish` networks with `depth_tau=3`, `depth_xi=3`, width 64, and $m = 3$ generators.

- **Example 4:** `tanh` MLPs with $\tau$ as $[1, 32, 32, 1]$ and $\xi$ as $[3, 64, 64, 2]$, with $m = 3$.

**Stage-2 loss terms and weighting.** The generator objective is a weighted sum of seven losses $L_1, \ldots, L_7$ computed on minibatches from $\mathcal{T}$. The code uses:

- **Structural/algebraic terms:** closure ($L_1$), Jacobi ($L_2$), skew-symmetry ($L_3$), bilinearity ($L_4$), and an independence/scaling term ($L_5$) with `s5_mode="sigma"` and `s5_tau=0.8` (Examples 1–4).

- **SDE-specific constraints:** the Ito determining equations ($L_6$) computed using surrogate coefficient functions $\mu(\cdot)$ and $\sigma(\cdot)$ derived from the learned Stage-1 network(s), and a pushforward consistency term ($L_7$) comparing transformed coefficients under the learned generator-induced flow, using a small finite $\epsilon$ step (e.g. `s7_eps=1e-2`) and typically a single step (e.g. `s7_steps=1` in the fixed-weight notebooks).

Weighting:

- **Examples 1–3 (fixed weights):** `LossWeights` sets

  $$(w_{1:7}) = (1.0, 0.1, 0.1, 0.1, \star, 1.0, 0.1),$$

  with $\star = 0.1$ in Examples 1 and 3, and $\star = 0.5$ in Example 2. Generator L2 decay is applied with `weight_decay=1e-6`.

- **Example 4 (scheduled weights):** $w_6 = 10$, $w_7 = 2$, and $w_5 = 2$ are kept large throughout, while $w_1 - w_4$ are ramped on only in the final 40% of training after 60% progress via a piecewise-linear schedule.

**Stage-2 optimization.**

- **Examples 1–3:** `optax.clip_by_global_norm (1.0) + optax.adam`. The nominal `GenTrainConfig.lr` is overridden to $10^{-4}$ in Examples 1 and 3; Example 4 uses $10^{-4}$ directly. Minibatches are sampled uniformly without replacement from the prepared `TX_gen` array after filtering non-finite rows.

- **Example 4:** `optax.clip_by_global_norm + optax.adamw` with a learning-rate schedule: 5% linear warmup from 0 to `lr`, followed by cosine decay down to `alpha=0.1` of `lr`. The sampler mixes half empirical `TX_gen` points and half uniform box points each step.

| Example | Steps | Batch | Optimizer | LR(s) | Weight decay | Extra regularizers |
|---|---|---|---|---|---|---|
| 1 | 10,000 | 4096 | Adam | $3 \times 10^{-3}$ | $10^{-6}$ (explicit L2) | — |
| 2 | 2,000 | 4096 | Adam | $3 \times 10^{-3}$ | $10^{-6}$ (explicit L2) | — |
| 3 | 10,000 | 4096 | Adam | $3 \times 10^{-3}$ | $10^{-6}$ (explicit L2) | — |
| 4 | 20,000 outer | 8192 | AdamW + clip (separate for drift/diff) | $2 \times 10^{-3}$ (drift), $10^{-3}$ (diff) | 0 | JVP-smooth $(10^{-3})$, var $(5 \times 10^{-4})$, whiten $(5 \times 10^{-3})$ |

*Table 13.* Stage-1 (surrogate) optimization settings. Example 4 alternates `drift_steps_per_iter=3` and `diff_steps_per_iter=1` inside each outer step.

| Example | $m$ | Stage-2 cloud source | $n_{\mathrm{traj}}$ (gen) | $B_{\mathrm{gen}}$ | $\tau$ net | $\xi$ net | Activation |
|---|---|---|---|---|---|---|---|
| 1 | 3 | surrogate sim, flatten $(t, x)$ | 256 | $256(500+1) = 128,256$ | $[1, 32, 32, 1]$ | $[2, 64, 64, 1]$ | tanh |
| 2 | 3 | surrogate sim + reflection + mixture cloud | 256 | kept at 128,256 | depth 3, width 64 | depth 3, width 64 | swish |
| 3 | 3 | surrogate sim, flatten $(t, x, y)$ | 256 | $256(200+1) = 51,456$ | $[1, 32, 32, 1]$ | $[3, 64, 64, 2]$ | tanh |
| 4 | 3 | data flatten $(t, x, y)$ | 2048 | $2048(200+1) = 411,648$ | $[1, 32, 32, 1]$ | $[3, 64, 64, 2]$ | tanh |

*Table 14.* Stage-2 (generator) architectures and the point clouds used for training the SDE symmetry generators. $B_{\mathrm{gen}}$ counts $(t, \text{state})$ points used for generator losses.

# Q. Sensitivity of the Maximum Principal Angle to Weight Choices

Table 16 reports the maximum principal angle as the weights $(w_6, w_7)$ are varied. Overall, the response is markedly non-monotone in both weights, indicating a strong interaction between the two penalty terms. Across the sweep, the maximum principal angle ranges from $14.71°$ (at $(w_6, w_7) = (1.0, 0.7)$) to $55.31°$ (at $(w_6, w_7) = (0.2, 0.9)$). For fixed $w_6$, changing $w_7$ can induce substantial variation: for example, at $w_6 = 0.2$ the angle increases from $21.35°$ at $w_7 = 0.1$ to $42.28°$ at $w_7 = 0.5$ and further to $55.31°$ at $w_7 = 0.9$, whereas at $w_6 = 0.6$ the angle peaks at $37.00°$ for $w_7 = 0.3$ and then decreases to $21.13°$ for $w_7 = 0.9$. Similarly, for fixed $w_7$, decreasing $w_6$ does not uniformly increase or decrease the angle (e.g., at $w_7 = 0.1$ the values span $22.64°$ at $w_6 = 1.0$ up to $39.21°$ at $w_6 = 0.4$ and down to $21.35°$ at $w_6 = 0.2$). These trends suggest that the maximum principal angle is sensitive to the relative balance between the two weights rather than to either weight in isolation, motivating the use of a small grid search over $(w_6, w_7)$ when selecting a stable operating point.

| Example | Steps | Batch | LR | Optimizer | Weights $(w_1, \ldots, w_7)$ | Sampling notes |
|---|---|---|---|---|---|---|
| 1 | 3000 | 2048 | $10^{-4}$ | clip(1.0)+Adam | (1,0.1,0.1,0.1,0.1,1,0.1) | uniform minibatches from `TX_gen` |
| 2 | 3000 | 1024 | $10^{-4}$ | clip(1.0)+Adam | (1,0.1,0.1,0.1,0.5,1,0.1) | $x$-reflection in sim; 70% uniform box points; optional traj balancing |
| 3 | 3000 | 1024 | $10^{-4}$ | clip(1.0)+Adam | (1,0.1,0.1,0.1,0.1,1,0.1) | uniform minibatches from `TX_gen`; `print_every=10` |
| 4 | 6000 | 256 | $2 \times 10^{-4}$ (scheduled) | clip(1.0)+AdamW | scheduled: $w_6{=}10, w_7{=}2, w_5{=}2$, ramp $w_{1:4}$ | half empirical `TX_gen` + half uniform box; warmup+cosine LR |

*Table 15.* Stage-2 (generator) optimization and weighting differences across examples.

*Table 16.* Maximum principal angle as a function of weights $w_6$ and $w_7$.

| $w_6$ | $w_7$ | Max. principal angle (deg) |
|---|---|---|
| 1.0 | 0.1 | 22.64 |
| 1.0 | 0.3 | 26.78 |
| 1.0 | 0.5 | 23.38 |
| 1.0 | 0.7 | 14.71 |
| 1.0 | 0.9 | 19.90 |
| 0.8 | 0.1 | 35.92 |
| 0.8 | 0.3 | 24.36 |
| 0.8 | 0.5 | 20.86 |
| 0.8 | 0.7 | 18.81 |
| 0.8 | 0.9 | 31.71 |
| 0.6 | 0.1 | 22.40 |
| 0.6 | 0.3 | 37.00 |
| 0.6 | 0.5 | 34.19 |
| 0.6 | 0.7 | 29.91 |
| 0.6 | 0.9 | 21.13 |
| 0.4 | 0.1 | 39.21 |
| 0.4 | 0.3 | 27.91 |
| 0.4 | 0.5 | 25.48 |
| 0.4 | 0.7 | 37.23 |
| 0.4 | 0.9 | 23.49 |
| 0.2 | 0.1 | 21.35 |
| 0.2 | 0.3 | 25.10 |
| 0.2 | 0.5 | 42.28 |
| 0.2 | 0.7 | 20.48 |
| 0.2 | 0.9 | 55.31 |

