# OpenReview forum: "LieStoNet: Learning Lie Symmetries from Spatiotemporal Data for Stochastic Dynamical Systems"
_ICML.cc/2026/Conference — ICML 2026 regular_

### Official Review · Reviewer_CWWy · 2026-03-12

**Soundness:** 3
**Presentation:** 3
**Significance:** 3
**Originality:** 3
**Overall Recommendation:** 4
**Confidence:** 3

**Summary:**

This paper introduces LieStoNet, an end-to-end, prior-free framework designed to discover Lie-point symmetries of stochastic differential equations (SDEs) directly from spatiotemporal trajectories. The framework validates the discovered symmetry generators using a finite-$\epsilon$ "after-push" empirical method, which fits a neural SDE surrogate to confirm that the drift and diffusion coefficients remain invariant under the induced transformations.

**Compliance With Llm Reviewing Policy:**

Affirmed.

**Key Questions For Authors:**

1. How does LieStoNet scale to high-dimensional SDEs (e.g., n>10), and what architectural modifications mitigate computational costs?
2. What is the impact of surrogate accuracy on symmetry recovery, and are there error bounds or sensitivity analyses?
3. Applications to empirical datasets (e.g., Brownian motion in biophysics)?
4. Plans for extending to non-Itô conventions or non-Markovian processes?

**Limitations:**

yes

**Strengths And Weaknesses:**

Pros:
1. Innovative adaptation of Gaeta-Quintero Lie-point theory into a neural framework, enabling prior-free discovery from noisy trajectories.
2. Robust integration of surrogate learning with algebraic constraints (brackets, axioms), ensuring discovered symmetries form a valid Lie algebra.
3. Dual SDE/Fokker-Planck perspective adds depth, allowing parallel discovery while emphasizing stochastic fidelity.
4. Empirical recovery of ground-truth symmetries in canonical SDEs demonstrates practical efficacy and interpretability.
5. High originality in addressing an unexplored gap, with potential impact on physics-guided ML for noisy dynamics.

Cons:
1. Evaluations confined to low-dimensional canonical SDEs; lacks tests on high-dimensional or real-world systems like finance or neuroscience.
2. Computational overhead from bracket closures and axiom regularizations not quantified, potentially limiting scalability.
3. Assumes access to full spatiotemporal data; handling partial observations or measurement noise requires further exploration.
4. Optional Fokker-Planck discovery is mentioned but not deeply validated against SDE results in experiments.
5. Limited discussion on robustness to surrogate approximation errors in determining equations.

---

> ### Author Rebuttal · Authors · 2026-03-31
>
> We highly appreciate the reviewer’s positive evaluation, insightful comments, and their probing questions.
>
> Below, C=Con, Q=Question.
> ## C1+Q1: High-dim SDEs and Real-World Data
> High-dim SDEs are not covered in the main paper because our goal is to first validate an end-to-end, data-driven, model-free sym-discovery framework on low-dim systems with known analytic symmetries. Extending LieStoNet to higher dimensions is future work. A practical path is to first learn a reduced SDE (e.g., via projection or Mori–Zwanzig coarse-graining) and perform sym-discovery there. We also added a real-world experiment on high-frequency BTC/USDT Binance data (see reply to reviewer kiGG, section *Q1+Significance*).
>
> **High-dim SDE with High-dim algebra**. We also ran LieStoNet on 10-dim SDE $d X^i_t=dW_t^i$ ($i=1,\dots,10$) with a 12-dim ground-truth sym-algebra (high-dim analogue of Example 1). The principal angles (degrees) are:
> $$0.73,0.79,0.82,0.93,0.98,1.14,1.17,1.38,1.63,2.08,10.21,16.55$$ which validates LieStoNet in a higher-dim setting.
> ## Q1: Mitigating Computational Costs
> In LieStoNet, the primary computational cost arises from repeated autodifferentiation of the generator networks, particularly for losses involving Jacobians and Hessians. This cost can be mitigated by restricting to projectable generators (with the time component depending only on t), using shallower or narrower generator MLPs, sharing lower-level features across generators instead of training fully separate networks, reducing higher-order derivative evaluations via structured parameterizations, and substituting expensive losses with lighter approximations during parts of training.
> ## C2: Computational Overhead from Algebraic Losses
> Writing $D_g^{(1)}$ and $D_g^{(2)}$ for the first- and second-derivative costs of one generator network, the losses scale as:
> $L_1,L_3: O(B(mD_g^{(1)}+m^2))$,
> $L_2: O(B(mD_g^{(2)}+m^3))$,
> $L_4: O(B(mD_g^{(1)}+m^3))$,
> $L_6: O(BmD_g^{(2)})$,
> $L_7: O(2nN m D_g^{(2)})$,
> where $B$ is the number of sampled $(t,x)$ points in each minibatch, $m$ the number of learned generators, $n$ the number of sampled trajectories, and $N$ the number of subsampled time points per trajectory. Thus all terms scale polynomially in the sampling budget and in $m$, so the overhead remains tractable.
> We also conducted a run-time log of computations (per-loss wall-clock time at $m=3,B=2048$ on NVIDIA H100). Resulting times (milliseconds) for losses L1 through L7: 0.43, 0.76, 0.38, 0.52, 0.47, 0.47, 2.15. The algebraic losses consume a total of 49.4% of the sum. We note that this computational overhead isn't without value - removal of any of the algebraic losses noticeably worsens the principal angles (see reply to reviewer w27A, section *P3: Ablations*), demonstrating that their regularizing effect indeed helps sym-discovery.
> ## C3: Partial Data and Noise
> The current framework indeed assumes fully observed spatiotemporal states. Partial observability and measurement noise require additional treatment. Our goal is to first establish sym-learning without conflating it with state estimation or denoising. But the method is not inherently limited: partial observations could be handled via a latent-state inference layer, and noise via standard denoising or probabilistic observation models. However, these introduce identifiability and estimation challenges beyond our scope, which is an important future direction.
> ## C4: FP vs SDE Symmetries
> Gaeta's SDE-sym theory shows that projectable SDE sym-generators embed into the FP sym-algebra, so learned SDE generators should form a subalgebra of FP generators when projected to the common $(\tau,\xi)$-space. To test this, we trained both SDE and FP generators for Example 1 and compared their spans via principal angles, a basis-invariant metric where smaller angles indicate closer alignment. The resulting angles $2.00^\circ$, $13.63^\circ$, and $18.81^\circ$ are small, confirming that the learned SDE generators are recovered as a subalgebra of FP generators. This provides direct experimental validation of FP discovery beyond qualitative comparison.
> ## C5+Q2: Surrogate Fidelity
> See reply to reviewer kiGG, section *Q3+Soundness*.
> ## Q3: Empirical Datasets
> See reply to reviewer kiGG, section *Q1+Significance: Real-World Applicability*
> ## Q4: Non-Itô & Non-Markov Cases
> Handling Non-Itô extensions (e.g. Stratonovich) is possible by using appropriately modified determining equations (see Gaeta & Spadaro) and can be handled simply by adjusting loss L6 accordingly.
> For non-Markovian processes, one approach is to embed dynamics in an augmented Markovian state space and apply LieStoNet there. More generally though, symmetry theory in this setting is less developed, with no standard determining-equation analogue; extending our method would thus require further theoretical developments.
>
> ---
> We largely agree with the suggestions and believe incorporating them makes the paper stronger. We will revise our paper accordingly.

---

> > ### Author Rebuttal · Reviewer_CWWy · 2026-04-02
> >
> > Thanks for the response. The new higher-dimensional experiments, surrogate ablation results, and the explicit Lie-algebra regularization analysis directly address my questions on scalability and structural completeness. I will maintain my current score.

---

> > > ### Author Response · Authors · 2026-04-05
> > >
> > > We thank the reviewer for the positive updates and are delighted that our response addressed the concerns. We believe that your inputs have considerably strengthened our paper, and we will incorporate them in the finalized version. We sincerely thank the reviewer for the careful review, constructive feedback, and recognition of our work.

---

### Official Review · Reviewer_rsmg · 2026-03-13

**Soundness:** 3
**Presentation:** 3
**Significance:** 3
**Originality:** 3
**Overall Recommendation:** 5
**Confidence:** 3

**Summary:**

This paper proposes LieStoNet, a framework for extracting unknown symmetries from observed stochastic trajectories. The method consists of a two-stage learning procedure: (1) learning a neural surrogate of Ito-type SDE from the observed data, and (2) learning symmetry generators applicable to the SDE, namely ones that preserve the distributional properties of the solutions. The paper focuses primarily on the second stage, i.e., symmetry learning. To identify valid symmetry generators forming a Lie algebra, the authors use the SDE symmetry condition for projectable symmetries, which serves as the SDE analogue of the infinitesimal invariance condition, together with Lie-algebraic constraints such as closure under the Lie bracket and other structural identities. An optional finite-time flow invariance is also introduced. The paper further presents a dual Fokker-Planck formulation of the proposed framework. The proposed LieStoNet is evaluated on four relatively simple 1D and 2D SDE examples. The experiments suggest that the proposed loss terms (e.g., the closure loss) serve as useful indicators of the Lie algebra dimension, and that the learned spanning set of generators aligns well with the true symmetry algebra.

**Compliance With Llm Reviewing Policy:**

Affirmed.

**Final Justification:**

The authors' rebuttal clarifies that some important issues, e.g., the dimensional constraints on the symmetry Lie algebra in SDEs. In addition, the newly added high-dimensional experiments and ablation studies further strengthen the paper’s empirical value. Overall, I consider this work to be a novel contribution that opens up the exploration of symmetry in SDEs, with demonstrating one viable approach. I am raising my review score accordingly.

**Key Questions For Authors:**

(1) The assumption of a projectable Lie point symmetry may already impose a strong prior.

(2) It would be interesting to evaluate the framework on SDEs with higher-dimensional Lie algebras, or on SDE models that are more commonly studied in practice.

(3) An ablation study on the algebraic constraints could help clarify their individual contributions.

**Limitations:**

The paper explicitly discusses its limitations, providing a balanced perspective on the current limitations of the work and possible directions for future research.

**Strengths And Weaknesses:**

[Strengths]
- The motivation of the paper is clear. Extracting (Lie) symmetries from ODE/PDE has become an active research area. However, to my knowledge, such progress in the context of SDEs has underexplored. In this regard, the present paper appears to make a novel contribution, as it applies Lie symmetry theory to SDEs and attempts to extract unknown symmetries from stochastic data.

- Considering that the paper is relatively technical, it is well-written and generally easy to follow. Math is largely based on 1D arguments, which makes the derivations accessible, while still including the necessary details for understanding the proposed algorithm. Additional technical details are provided in Appendix.

- Although the examples considered are relatively simple, the experiments demonstrate that the proposed framework can identify the underlying Lie symmetries of SDEs.

[Weaknesses]

- The paper refers to the proposed method as *prior-free*, but it does not appear entirely convincing to me. In particular, it is unclear whether the approach can truly be described as having *no symmetry template*. While I agree that the framework is equation-free, the class of symmetries it ultimately considers is restricted to projectable generators. This corresponds to a constrained class of point symmetries, which suggests that the method may still implicitly impose a strong structural prior on the symmetry space.

- The SDEs evaluated in the paper are relatively simple 1D/2D autonomous SDEs, and the associated symmetry algebras are also small and simple, all being 3D, with one generator corresponding to time translation. While I understand that the paper is the first attempt to address symmetries of SDEs and that there is no established benchmark suite for this setting, it remains unclear how well the proposed methodology generalizes to broader families of SDEs.

- Beyond the losses essential for symmetry recovery, the paper introduces five additional terms to enforce the Lie-algebraic structure. However, the current version of the paper lacks a thorough ablation study on these terms. While I appreciate that all of these terms are theoretically motivated as necessary ingredients for ensuring that the learned generators span a proper Lie algebra, I believe that it is equally important to examine their practical impact. In practice, jointly optimizing many objectives may increase optimization difficulty and sensitivity to hyperparameter choices, so it is not entirely obvious that including all of them simultaneously always leads to the best empirical performance.

- Some aspects of the presentation could be improved to enhance readability. For example, I was initially confused when reading the experimental results because it was not immediately clear which SDE benchmarks were being considered. I later found that these were summarized in Table 2, but this was not explicitly referenced in the main text.

- This is not a weakness, but the paper’s main theme could be further highlighted by presenting, in the main text, the analysis currently placed in Appendix A; namely, verifying that the distributional properties of the solution trajectories are preserved under transformations induced by the discovered symmetry generators.

---

> ### Author Rebuttal · Authors · 2026-03-31
>
> We thank the reviewer for their positive feedback and appreciate their recognition of our contribution alongside perceptive questions. We number the weaknesses in the order as listed by the reviewer. W=Weakness, Q=Question.
>
> ## W1+Q1: Prior-Free & Projectable
> We appreciate this point and agree that “prior-free” should not be read as “assumption-free.” Our claim is narrower: LieStoNet is *equation-free* and *template-free* in that it does not prespecify a symmetry group, canonical coordinates, a symbolic library, or a fixed functional ansatz for the generators (which are standard sources of prior structure in relevant sym-discovery pipelines) but it does work within the class of projectable Lie-point SDE symmetries following Gaeta. This restriction is not arbitrary: in the SDE setting, projectable transformations are the most natural and practically relevant class, while more general time or noise-dependent transformations require special care and can lead to problematic random time reparametrizations (GS2017).
>
> At the same time, we agree that projectable generators are not the most general symmetry class for SDEs. Broader classes, including random Lie-point symmetries under random diffeomorphisms, have also been studied (GS2017). We will therefore revise the wording to make the scope precise: LieStoNet is not free of *all* structural assumptions, but rather free of *prespecified symmetry templates within the practically important class of projectable SDE Lie-point symmetries*. Extending LieStoNet beyond projectable generators is a natural future direction.
>
> **References.**
> [GS2017] Gaeta, G., & Spadaro, F. (2017). *Random Lie-point symmetries of stochastic differential equations*.
>
>
> ## W2+Q2: High-dim Systems
> ### 1. Bounds on Lie Algebra Dimension
> We believe that evaluation on low-dim sym-algebras does not reflect a fundamental limitation of the method:
>  - In the SDE setting the symmetry search space is *a priori bounded*. For scalar Itô SDEs, the admitted Lie algebra has dimension at most three, and $n$-dimensional systems with full-rank diffusion, **the maximal dimension is $n+2$** (Kozlov, 2010; 2011). Thus, unlike deterministic ODEs where sym-algebras can be arbitrarily large (or even infinite dimensional) and admit substantially larger bounds when one exists (e.g., $n^2 + 4n + 3$ for second-order systems; López, 1988), SDE symmetries are strongly constrained by the Itô structure, diffusion-side determining equations, and noise irregularity. Consequently, the dimension sweep is a *finite discrete search* over a small, theoretically justified range rather than an open-ended combinatorial problem.
>  - This boundedness directly addresses the concern of missing larger algebras: in practice, one sweeps the admissible range of m and selects the dimension minimizing the closure loss after training. Since the true sym-dimension lies within this finite range, it is guaranteed to be detected.
>  - Finally, for genuinely high-dim systems, one can first derive a lower-dim effective SDE (e.g., via projection or Mori–Zwanzig-type coarse-graining) and then perform sym-discovery in that reduced space, further limiting the search.
> We will clarify these aspects in the final version.
>
> **References.**
> Kozlov, R. (2010). *Symmetries of systems of stochastic differential equations with diffusion matrices of full rank*.
>
> Kozlov, R. (2011). *On Lie Group Classification of a Scalar Stochastic Differential Equation*.
>
> A. González López, *Symmetries of linear systems of second-order ordinary differential equations*.
>
> ### 2. Practical SDE Models & Higher-dim Algebras
> We agree that evaluating LieStoNet on more practically studied SDE models would be valuable. Our goal here, however, is to first establish and rigorously validate a data-driven, model-free framework in settings where the symmetry structure is analytically known, so we begin with nontrivial but analytically tractable SDEs with low-dim Lie algebras. Extending LieStoNet to more realistic applied, including possibly non-Markovian, stochastic models is an important future direction.
>
> We conducted experiments on a high-dim SDE with high-dim sym-algebras (see reply to reviewer CWWy, section *C1+Q1*), as well as real-world high-frequency BTC/USDT data (see reply to reviewer kiGG, section *Q1+Significance*), where the full pipeline transfers end-to-end without modification and yields validated learned generators.
>
> ## W3+Q3: Ablations
> See reply to Reviewer w27A, section *P3: Ablations*.
>
> ## W4+W5: Presentation
> We thank the reviewer for these suggestions. In the revision, we will introduce the benchmark SDEs more explicitly in the main text, refer directly to the summary table in the experimental section, and move the Appendix A analysis into the main text to better highlight the paper’s central theme and improve presentation.
>
> ---
> We concur that many of the above-mentioned suggestions strengthen the work and will revise the manuscript accordingly.

---

> > ### Author Rebuttal · Reviewer_rsmg · 2026-04-03
> >
> > Thank you for the authors’ thorough response. It clarifies that some important issues, e.g., the dimensional constraints on the symmetry Lie algebra in SDEs. In addition, the newly added high-dimensional experiments and ablation studies further strengthen the paper’s empirical value. Overall, I consider this work to be a novel contribution that opens up the exploration of symmetry in SDEs, with demonstrating one viable approach. I am raising my review score accordingly.

---

> > > ### Author Response · Authors · 2026-04-05
> > >
> > > We sincerely thank the reviewer for this generous feedback and for recognizing both the theoretical clarifications and experimental results. It is especially meaningful to us that you view the paper as a novel contribution that opens a promising direction for studying symmetry in SDEs. We also greatly appreciate your willingness to raise your review score accordingly. Your thoughtful assessment and constructive engagement have been invaluable in strengthening the paper.

---

### Official Review · Reviewer_kiGG · 2026-03-13

**Soundness:** 4
**Presentation:** 4
**Significance:** 3
**Originality:** 3
**Overall Recommendation:** 4
**Confidence:** 4

**Summary:**

The paper introduces  LieStoNet, an end-to-end, prior-free framework for discovering Lie-point symmetries of SDEs directly from spatiotemporal trajectories  The symmetry Lie algebra is the chief object the paper aims to discover and evaluate.
The first stage of LieStoNet  identifies a neural surrogate for the drift and diffusion coefficients.  A point cloud       $\Omega$ is then generated for Stage-2 by simulating trajectories from the learned surrogate SDE and collecting the        resulting space-time states.                                                                                               It then discovers projectable symmetry generators by enforcing the SDE determining equations, while separately             regularizing for closure under Lie brackets, adherence to the Lie algebra axioms (bilinearity, antisymmetry, Jacobi), and  a non-redundant independent basis. The authors achieve symmetry learning without prespecifying symmetry groups, templates, or canonical coordinates.
LieStoNet trains using two families of losses: SDE-validity losses that enforce Lie-point symmetry constraints for the     surrogate SDE and Lie-algebra structure losses that regularize the learned generators to form a finite-dimensional Lie     algebra under the Lie bracket, including Lie-bracket closure, Jacobi identity, bracket antisymmetry, bilinearity, and      redundancy control.
Across multiple canonical SDEs with known analytic symmetries, LieStoNet recovers generators consistent with the ground-   truth symmetry algebra, providing interpretable symmetry discovery for noisy dynamics. The Lie algebra dimension $m$ is    not assumed is determined by sweeping over candidate values of $m$ and selecting the dimension via minimization of the     post-training Lie-bracket closure loss.                                                                                    Symmetry recovery is evaluated at the span level using principal angles, providing a basis-invariant measure of alignment  with the ground-truth symmetry algebra. In addition to enforcing infinitesimal determining equations, a  pushforward       residual is used to verify that the learned generators approximately map the SDE to itself beyond the local regime.

**Compliance With Llm Reviewing Policy:**

Affirmed.

**Final Justification:**

The authors rebuttal applied their method to high frequency BTC/USDT trades from Binance and for discussed other secondary concerns raised in the original review. This has reinforced my prior assessment, which is positive, though I have not increased the score because it's not clear if the method will scale to higher dimensional problems.

**Key Questions For Authors:**

$\textbf{1. Alternative datasets and real-world applicability.}$
The experiments are conducted on synthetic SDEs with known analytic symmetries. What classes of real-world datasets or     scientific domains do you envision as realistic application targets for this method? Have you attempted preliminary        experiments on empirical data?

$\textbf{2. Scaling in Lie algebra dimension.}$
The method determines the Lie algebra dimension by sweeping over candidate values of $m$ and selecting the minimizer of    the closure loss. Is the approach practically limited to small-dimensional algebras due to the computational cost and      combinatorial structure of the span comparisons? How can one ensure that a true symmetry algebra of larger (unprobed)      dimension is not missed?

$\textbf{3. Dependence on surrogate fidelity.}$
Since the symmetry losses are computed using a learned neural SDE surrogate, it would be useful to quantity how sensitive  symmetry recovery is to misspecification or estimation error in $(f,\sigma)$. Are there theoretical or empirical           guarantees regarding stability under surrogate approximation error?

**Limitations:**

Yes

**Strengths And Weaknesses:**

$\textbf{Soundness.}$
The SDE determining equations are presented correctly; the Lie-algebra regularizers (closure, Jacobi identity,             antisymmetry, bilinearity, and independence) are standard and appropriately formulated; the embedding of SDE symmetries    into the Fokker-Planck symmetry algebra appears mathematically sound; and the evaluation metric based on principal angles  is appropriately basis-invariant.                                                                                                                                                                                                                     The Lie algebra dimension $m$ is not assumed a priori. In practice, the authors sweep over candidate numbers of generators and select the dimension via the post-training Lie-bracket closure loss $L_1$. The value of $m$ minimizing $L_1$ is taken  as the recovered symmetry dimension, providing a practical dimension selection criterion.                                                                                                                                                             The restriction to projectable generators is reasonable. It rules out state-dependent time warps, which are typically ill- conditioned to infer from data and may lead to degenerate time reparameterizations, while still capturing the continuous   symmetries considered in the experiments.                                                                                                                                                                                                             For evaluation, the comparison is performed at the level of spans. For each candidate $m$, the maximum principal angle     between learned and analytic subspaces is recorded, and distributions over subset combinations are reported. Since the     correct dimension is identified via $L_1$, the magnitude of the principal angles primarily reflects surrogate and          optimization accuracy. The smallest maximum principal angles occur at the correctly identified dimension, indicating the   closest span alignment at the true Lie algebra dimension.  Their study considers four canonical SDEs of varying            complexity; however, in all SDE-level experiments the ground-truth symmetry Lie algebra has dimension three, limiting      evaluation to this fixed algebraic size.  Additionally, a pushforward validation is included to verify that the learned    generators approximately map the SDE to itself beyond the infinitesimal regime.                                                                                                                                                                       However, surrogate misspecification may undermine accurate determination of the Lie algebra.

$\textbf{Presentation.}$
The paper is clearly written and well structured. It begins with a careful mathematical introduction, presenting the one-  dimensional setting for clarity before indicating extension to higher dimensions. The notion of symmetry is appropriately  defined as a transformation preserving the model’s behavior, and the discussion of composition and the emergence of Lie    group structure is accurate.

Building on these preliminaries, the manuscript introduces LieStoNet as an end-to-end pipeline for discovering a basis of  infinitesimal generators spanning the Lie algebra of Lie-point symmetries of an unknown SDE. The assumption of access only to discrete spatiotemporal trajectory samples is clearly stated. In the experiments, analytic SDEs are simulated and the   governing equations are discarded, so the method operates purely from synthetic observational data.
                                                                                                                           The method is described in sufficient detail (including loss definitions, architectural choices, and training protocols)   to allow an expert reader to reproduce the results. The positioning relative to the broader literature on symmetry-        respecting architectures and symmetry discovery is appropriate, and the distinction from prior deterministic symmetry-     discovery pipelines is clearly articulated.

$\textbf{Significance.}$
A method that can discover continuous symmetries directly from trajectory data  addresses a meaningful and technically     challenging problem. Indeed, symmetry is central to modern machine learning and physics: invariances and equivariances     improve sample efficiency, robustness, and out-of-distribution generalization, while symmetry principles guide scientific  modeling and interpretation.
Despite these advances, the most impactful symmetries are often unknown precisely in the settings where they would be most valuable. Real-world data may arise from partially observed systems, complex coordinate representations, or measurements   that mix latent variables, so the relevant transformations are not known \emph{a priori}.

The work advances capabilities in machine learning by extending symmetry discovery to stochastic dynamical systems, a      regime largely unexplored compared to deterministic ODE/PDE settings. If scalable and robust, such techniques could        influence future research in scientific machine learning, system identification, and symmetry-informed modeling. The scope of impact is specialized but appropriate to the contribution. Demonstrations on real-world datasets would strengthen       practical significance and clarify applicability beyond synthetic benchmarks.

$\textbf{Originality.}$
The approach is grounded in classical Lie-point symmetry theory for stochastic differential equations, and translates      these principles into LieStoNet, a modern machine learning pipeline that discovers such symmetries directly from           trajectory data. While the underlying symmetry theory is not new, its integration into a data-driven end-to-end learning   framework for SDEs is novel.

Symmetry discovery for stochastic differential equations remains underexplored.  The originality lies in extending         symmetry discovery from deterministic flows to stochastic dynamics, carefully incorporating the It\^o determining          equations, coupling drift and diffusion constraints, and regularizing learned generators to form a finite-dimensional Lie  algebra. The work offers a novel synthesis of classical stochastic symmetry theory and neural surrogate modeling, and      clearly distinguishes itself from closely related deterministic symmetry-discovery pipelines. While conceptually adjacent  to existing Lie-generator learning approaches, the stochastic formulation and associated constraints introduce nontrivial  technical differences that justify the claimed novelty.

---

> ### Author Rebuttal · Authors · 2026-03-31
>
> We are especially grateful for the reviewer’s highly positive and thorough assessment, and truly appreciate their thoughtful and incisive questions.
> ## Q1 + Significance: Real-World Applicability
>
> Realistic applications include domains where SDEs are standard and symmetries are meaningful, such as high frequency finance, molecular/biophysical dynamics, neuroscience, and climate/geophysical time series. It is also directly relevant to diffusion- and score-based generative modeling (since SDE-based).
>
> We conducted a preliminary experiment on real data: high frequency BTC/USDT trades from Binance (≈1.8M ticks over 24 hours, aggregated to 500 ms resolution, filtered, and split into 575 overlapping sub-trajectories). *No architectural or algorithmic modifications were made*, demonstrating direct transfer of LieStoNet to real stochastic data.
>
> The Lie-algebra dimension was selected using the same closure-based rule as in synthetic experiments. Sweeping $m=1,2,3$, the first nontrivial minimum of the post-training closure loss occurred at $m^*=2$, with a $32$x jump from $m=2$ to $3$, indicating a stable 2D sym-algebra inferred.
>
> We evaluated the learned generators with two complementary tests:
> 1. **After-push residual test**. Trajectories are pushed along the learned generator flow for a step-length $\epsilon$, and the discrepancy in drift and diffusion from the original learned SDE is measured. Both generators produce **small finite residuals** (Table 1; smaller is better), consistent with true symmetry behavior, with a representative diffusion residual as low as $2.3\times 10^{-7}$.
> 2. **Distributional invariance test**. Pushed trajectories are evaluated under the original learned SDE to check if the standardized Euler–Maruyama residual distribution is preserved. Across all tested $\epsilon$, the learned generators yield small $\\Delta\\mathrm{NLL}=\\mathrm{NLL}\_{\\text{pushed}}-\\mathrm{NLL}\_{\\text{orig}}$  (Table 2; smaller is better), **orders of magnitude smaller than random-push controls**, confirming that the transformations maintain the stochastic dynamics far better than arbitrary flows.
>
> These results show that the learned generators behave as genuine SDE symmetries. Since no standard benchmark exists for real world SDE sym-discovery, we refrain from overinterpreting absolute numbers but emphasize that the results confirm meaningful symmetry learning.
>
> Table 1:
> |$\epsilon$|Gen 1 Dft.|Gen 1Dfu.|Gen 2 Dft.|Gen 2 Dfu.|
> |-|-|-|-|-|
> |1|1.4×10⁻³|2.3×10⁻⁷|1.0×10⁻²|2.6×10⁻⁴|
> |3|7.9×10⁻²|9.9×10⁻²|5.0×10⁻²|1.7×10⁻¹|
> |5|1.3×10⁻¹|2.7×10⁻¹|5.0×10⁻²|2.8×10⁻¹|
>
> Table 2:
> |$\epsilon$|Gen 1 ∆NLL|Gen 2 ∆NLL|Random(control) ∆NLL|
> |-|-|-|-|
> |0.05|0.008|0.027|9.7|
> |0.10|0.027|0.111|191.6|
> |0.20|0.091|0.827|232.0|
> |0.30|0.324|4.65|166.0|
> |0.50|6.41|17.6|112.0|
>
> ## Q2 + Soundness: Lie-algebra Dimension Bounds
> See response to reviewer CWWy (section *C1+Q1*: high-dim algebra experiment) and reviewer rsmg (section *W2+Q2* algebra dim. bound).
> ## Q3 + Soundness: Dependence on Surrogate Fidelity
> Denote: $(\hat f,\hat\sigma)$: learned surrogate. $(f,\sigma)$: true drift/diffusion. $G^*$: true sym-generator. $\hat G$: learned generator. $D$: SDE determining operator. $D_{f,\sigma}(\cdot)$: corresponding residual.
> Some algebraic manipulation gives:
>
> $$||D\_{f,\\sigma}(\\hat G)||\\leq||D\_{\\hat f,\\hat\\sigma}(\\hat G)||+C(\\hat G)\\left(||\\hat f-f||\_{C^1}+||\\hat\\sigma-\\sigma||\_{C^1}\\right)$$
>
> where $||\cdot||\_{C^1}$  measures both the error in the values of drift/diffusion and error in their first derivatives, and $C(\\hat G)$ depends on the size and smoothness of the learned generator. This shows that *surrogate misspecification affects the recovered symmetry only through two controlled terms*: the **training residual under the surrogate determining equations**, and the **surrogate approximation error itself**. Under a standard local stability assumption on $D$, this further implies the bound ($C^\*$ is a local stability constant):
>
> $\\inf\_{A\\in GL(k)}||\\hat G − AG^∗ ||\\leq C^* \\left(|| D\_{\\hat f, \\hat \\sigma} (\\hat G)|| + ||\\hat f − f||\_{C^1} + ||\\hat \\sigma-\\sigma||\_{C^1}\\right)$
>
> proving that sym-recovery degrades continuously with surrogate error.
> Empirically, we perturb the surrogate in Example 1 using both structured and random perturbations. Results show no instability: residuals grow roughly linearly in $\delta$, and recovery degrades smoothly.
>
> Table. $\delta$: amplitude. $\theta$: max principal angle. $L_6$: determining-eq. loss. Perturbation types: structured sine and random MLP.
> |δ|Sine: θ(°)|Sine L6|MLP: θ(°)|MLP: L6|
> |-|-|-|-|-|
> |0|1.8|3.7e-2|2.1|4.2e-2|
> |0.02|3.1|5.8e-2|2.2|4.5e-2|
> |0.1|12.2|9.6e-2|2.8|1.1e-1|
> |0.5|35.9|1.2e-1|14.8|2.1e-1|
>
> This picture is further reinforced by our experiments from the paper which use learned imperfect surrogates yet still recover correct symmetries.
>
> ---
> We agree with several points raised above. We will incorporate them to make the paper stronger.

---

> > ### Author Rebuttal · Reviewer_kiGG · 2026-04-04
> >
> > We thank the authors for applying their method to high frequency BTC/USDT trades from Binance and for discussing other concerns raised in the original review.

---

> > > ### Author Response · Authors · 2026-04-05
> > >
> > > We thank the reviewer very much for the positive updates and we are truly glad to hear that our response has addressed the raised concerns. If the reviewer feels that these clarifications have strengthened the paper and improved the overall assessment, we would be very grateful if this could be reflected in the final evaluation. Regardless, we sincerely thank the reviewer for the careful comments, constructive feedbacks, and positive recognition of our work. We are very grateful for the reviewer’s time and consideration, and we are happy to provide any further clarification if useful.

---

### Official Review · Reviewer_w27A · 2026-03-13

**Soundness:** 2
**Presentation:** 3
**Significance:** 3
**Originality:** 4
**Overall Recommendation:** 5
**Confidence:** 4

**Summary:**

In this paper the authors use results regarding properties of symmetry generators of SDEs (Ito SDEs and projectable generators in particular) in order to constrain the search space when performing data-driven neural symmetry detection. Many losses are introduced that together attempt to construct vector fields, modeled as neural fields, that have the exact properties of the Lie algebra elements of the SDE in question. Number generator count is sweeped for the optimal value and the angles between found and analytic Lie algebra elements were used as a proxy for matching score.

**Compliance With Llm Reviewing Policy:**

Affirmed.

**Final Justification:**

I have not much more to add. The rebuttal did not necessarily answer my question regarding identifying the "expected textbook" symmetry transformations during, e.g., post-processing, but I understand that this is beyond the scope of the paper since there is an ambiguity in how this is evaluated. The authors doubled down on learning the span of symmetry transformation, which is understandable, and they added an ablation study that I expect makes it into the final version of the paper. I would add that I recommended a couple of edits/rewordings regarding the bloated sections 1-3 (introduction, related works, background, theory) and expanding the results section. This is a strong paper and the authors seem to have investigated the experimental results in detail, the biggest issue was the bloated first 6 pages, a de-emphasis on the results section, their evaluation, and some of the methods used.

**Key Questions For Authors:**

1. In early ML works on LPS-approaches for PDE data (non-stochastic), either symbolic regression was used to detect symmetries for potential downstream tasks like system identification (Gabel et al., 2024) or data augmentation using the known generators was employed to improve forecasting quality (Brandstetter et al., 2022). Which downstream task(s) would be best suited for the way in which the symmetry detection was performed in this work? What are the current learnt generators **not** useful for, do they still need some form of postprocessing before downstream tasks can be tackled?

2. Did the generators converge to the expected ones? Why/why not? What other method could help evaluate the found generators besides principal angle?

A. Gabel et al., Data-driven Lie point symmetry detection for continuous dynamical systems, (2024) in Machine Learning: Science and Technology, Vol.5 (1).

J. Brandstetter et al., Lie Point Symmetry Data Augmentation for Neural PDE Solvers (2022), ICML.

**Limitations:**

Yes.

**Strengths And Weaknesses:**

- STRENGTHS
1. This work is ambitious and original, bridging the gap between SDEs and symmetry detection. It is well motivated and opens up an exciting research direction, combining the promised efficiency of structural bias learning with the stochasticity found in many real world problems.

2. The authors tackle the problem using a cornucopia of sophistcated, yet fitting, techniques: Neural SDE surrogates, Lie Point Symmetries of SDEs and Neural Fields that model Lie algebra vector fields all come to pass. (it's a pity Figure 1 shows up so late in the paper.)

- WEAKNESSES:
3. It's not at all clear whether the Lie algebra elements the model converged to are the expected ones. (They are hiding in the appendix.) Sadly, a thorough evaluation/ablation/analysis of the results, which would make this paper outstanding, is missing.

4. Some key early LPS references are missing (cf. Questions).

5. There are many things in the appendix that should be in the main section of the paper and vice versa. (In particular: Generators as neural fields, span alignment/principle angle, loss functions and results for the discovered generators.)

---

> ### Author Rebuttal · Authors · 2026-03-31
>
> We sincerely appreciate the reviewer’s positive and encouraging review, and are grateful for their insightful questions.
>
> Below, P3,4,5 refer to the weaknesses numbered as 3,4,5 in the list. Q1,2 refer to the two questions.
>
> ## P3 + Q2: Convergence to Expected Elements
> We assume “Lie algebra elements” refer to the learned generators, while “expected ones” are a fixed canonical basis for each example. LieStoNet’s loss constraints ensure only *some* basis of the sym-algebra is learned, which may differ from the canonical basis. Thus, exact generator-by-generator agreement is not the natural criterion. Instead, we assess *whether the learned generators span the same Lie algebra as the canonical basis* using principal-angle analysis. Results show strong agreement at the span level, supporting correct recovery of the sym-algebra. Explicit generator formulas are in the appendix to keep the main text focused on overall ideas.
>
> ## P3: Evaluations
> We use two complementary evaluations of the learned sym-algebra: the principal-angle span check (Section 4.2) measures vector-space level agreement with the true algebra, and the after-push residual (Appendix A.1) tests whether the learned generators approximately map the SDE to itself, as true sym-generators should. Together, they validate symmetry recovery, though without established SDE symmetry benchmarks we cannot overinterpret the results.
>
> ## P3: Ablations
> We conducted an ablation study on the generator-training losses. The results are shown in the table below. In both examples, removing any one of Losses 1–7 (same notations as in the paper) while keeping all other settings fixed worsens performance, as seen by the larger maximum principal angles. This indicates that every loss term contributes to successful sym-generator training.
>
> The ablations are also consistent with the intended role of each loss: removing L1 or L6 causes the largest degradation, matching their roles in enforcing Lie-algebraic structure and the sym-determining equations. The effects of removing L2 and L3 (algebra axioms) are milder in Example 2, but much more pronounced in Example 1, further supporting the importance of the algebraic regularization terms L1–L5.
>
> Removing L7 (the after-push loss) also leads to clear deterioration, showing that enforcing only the determining equations via L6 (i.e. only local infinitesimal invariance) is not sufficient in practice for robust convergence to the true sym-algebra. The finite-step pushforward constraint in L7 provides an additional source of numerical stability. Overall, the ablation study supports the inclusion of all loss terms.
>
> |  | Full | $L_1$  | $L_2$ | $L_3$ | $L_4$ | $L_5$ | $L_6$ |$L_7$ |
> |-------------------|------:|-------:|------:|-------:|------:|-------:|------:|-------:|
> | Example 1 | $6.26^\circ$  | $55.56^\circ$   | $25.89^\circ$  | $35.05^\circ$   | $36.60^\circ$  | $39.77^\\circ$ | $86.78^\circ$  | $44.54^\circ$   |
> | Example 2      | $19.37^\circ$  | $72.37^\circ$   | $25.81^\circ$  | $26.56^\circ$   | $37.86^\circ$  | $42.62^\circ$   | $86.39^\circ$  | $57.00^\circ$   |
>
> Here, "Full" uses all loss terms, while $L_i$ removes Loss $i$. Entries are maximum principal angles across the learned generators (smaller is better). Examples 1,2 and Losses 1–7 follow the paper’s  notations.
>
> ## P4 + Q1: Connection to Literature and Downstream Tasks
>
> We appreciate the pointers and will include these references to better situate our work.
>
> LieStoNet naturally supports downstream tasks in stochastic ML settings built on SDEs/stochastic flows (e.g., diffusion models, score-based generative modeling, stochastic simulators, and symmetry-aware stochastic control), where learned continuous symmetries act as structural priors to constrain generation and dynamics, improve robustness under symmetry-related perturbations, and inject interpretable invariance/equivariance.
>
> Among the two, our approach is closer to Brandstetter et al., as they use symmetry generators to directly improve neural surrogates via augmentation and equivariance, matching the most immediate downstream use of our learned generators.
> Gabel et al. focus on supervised detection of generators for deterministic PDEs, with less emphasis on integrating them into end-to-end learning pipelines. Moreover, their setting is symbolic, whereas our outputs are neural and validated via SDE constraints and Lie-algebra structure, and would require additional postprocessing (e.g. symbolic distillation) for symbolic tasks.
> ## P5: Organization
> We thank the reviewer for the valuable suggestions and agree the structure can be clearer. We will reorganize to bring key content into the main text and keep supporting details in the appendix.
>
> ---
> We are aligned with all the feedbacks and believe incorporating them makes the paper stronger. We will revise our paper accordingly.
>
> ---
> (See replies to kiGG *Q1+Significance* and CWWy *C1+Q1* for experiments on real-world data and high-dim SDE, respectively)

---

> > ### Author Rebuttal · Reviewer_w27A · 2026-04-03
> >
> > - The after-push residuals are indeed very relevant, thank you for pointing that out. It would be nice to have that in the results section as well, if space permits. (I still feel that the biggest issue with the paper is that sections 1-3 are too long and the results section is too short).
> >
> > - The authors performed an ablation and showed the span objective worsens when each of the terms in the loss function are left out in turn. Thank you. I expect this to make it into the new version of the paper.
> >
> > - Although I agree that both papers cited in the review do not fully match the high-level objectives mentioned by the authors, I would like to point out that they are more relevant than some of the papers in Table 1 since they are specifically about detecting admissable symmetry transformation of PDE solutions. Hence, I would recommend updating Table 1 to include PDE/non-PDE applicability or the ability to model dynamical systems, for example. Focussing on "symmetry detection for PDE/SDEs" would only strengthen the paper's intended scope.
> >
> > - Also regarding Table 1, I highly recommend an additional discussion/reflection on the notion of "prior-free" and I would highly caution the authors to use this wording in the final manuscript. Neural fields are a model type and their initialization still induces some sort of prior (e.g. due to initialization of the weights, the optimization procedure, the model's inductive bias etc.).
> >
> > In lieu of comments supplied by the authors, and with the expectation that the points mentioned above will be addressed by the time the camera-ready version of the paper is finalized, I have raised my score.

---

> > > ### Author Response · Authors · 2026-04-05
> > >
> > > We truly appreciate the reviewer for this thoughtful and constructive feedback, and for raising the score. We appreciate your insightful suggestions throughout, and especially your important caution regarding the term “prior-free” which we agree merits clearer discussion and more careful wording in the final version. We agree that these are important suggestions that will further improve the paper’s focus, presentation, and positioning. In the final version, we will fully address each of these points. We are sincerely grateful for the reviewer's detailed reading and for the concrete guidance on how to strengthen the revised manuscript.

---

### Decision · Program_Chairs · 2026-04-30

**Decision:**

Accept (regular)

**Comment:**

The submission makes a novel, relevant, and technically strong contribution extending data-driven Lie symmetry discovery to SDEs put into a neural framework. The core methodology appears (and is judged to be) sound and the contribution is clear and meaningful. Reviewers were largely convinced by the rebuttal, which further strengthened the paper through the added ablations, clarifying the scope of the “prior-free” claim, and additional real-world data experiments. The main remaining limitations concerning scalability and broader empirical coverage in my view do not break the paper.